**Solar radiation estimation in West Africa: impact of dust conditions during**
**2021 dry season**
Léo Clauzel[1], Sandrine Anquetin[1], Christophe Lavaysse[1], Gilles Bergametti[2], Christel
Bouet[2,3], Guillaume Siour[4], Rémy Lapere[1], Béatrice Marticorena[4], Jennie Thomas[1]
[1]Université Grenoble Alpes, IRD, CNRS, Grenoble-INP, IGE, 38000 Grenoble, France
[2]LISA, Université Paris Cité and Univ Paris Est Créteil, CNRS, F-75013 Paris, France
[3]Institut d'Ecologie et des Sciences de l'Environnement de Paris, UMR IRD 242, Univ Paris
Est Créteil–Sorbonne Université–CNRS–INRAE–Université Paris Cité, F-93143 Bondy,
France
[4]LISA, Univ Paris Est Créteil, Université Paris Cité, CNRS, LISA, F-94010 Créteil, France
*Correspondence to:* Léo Clauzel (leo.clauzel@univ-grenoble-alpes.fr)

**Abstract**

The anticipated increase in solar energy production in West Africa requires high-quality solar irradiance estimates, which is affected by meteorological conditions and in particular the presence of desert dust aerosols. This study examines the impact of incorporating desert dust into solar irradiance and surface temperature estimations. The research focuses on a case study of a dust event in March 2021, which is characteristic of the dry season in West Africa. Significant desert aerosol emissions at the Bodélé depression are associated with a Harmattan flow that transports the plume westwards. Simulations of this dust event were conducted using the WRF meteorological model alone, as well as coupled with the CHIMERE chemistry-transport model, using three different datasets for the dust aerosol initial and boundary conditions (CAMS, GOCART, MERRA2). Results show that considering desert dust reduces estimation errors in global horizontal irradiance (GHI) by about 75%. The dust plume caused an average 18% reduction in surface solar irradiance during the event. Additionally, the simulations indicated a positive bias in aerosol optical depth (AOD) and PM10 surface concentrations. The choice of dataset for initial and boundary conditions minimally influenced GHI, surface temperature, and AOD estimates, whereas PM10 concentrations and aerosol size distribution were significantly affected. This study underscores the importance of incorporating dust aerosols into solar forecasting for better accuracy.

**Short summary**

Solar energy production in West Africa is set to rise, needing accurate solar irradiance estimates, which is affected by desert dust. This work analyses a March 2021 dust event using a modelling strategy  incorporating desert dust. Results show that considering desert dust cut errors in solar irradiance estimates by 75% and reduces surface solar radiation by 18%. This highlights the importance of incorporating dust aerosols into solar forecasting for better accuracy.

## 1. Introduction

The West African region is facing significant development challenges due to global change. One of these challenges is related to access to electricity, particularly through the use of renewable energy. West African countries have committed to reduce their greenhouse gas emissions as part of the Paris Agreement (2015). Furthermore, assessments of solar resources in West Africa demonstrate the region's substantial potential, as shown by Diabaté et al. (2004), Plain et al. (2019) and Yushchenko al. (2018). The International Energy Agency (IEA) projects that the installed capacity for photovoltaic (PV) power generation will increase by almost 20 times from 2020 to 2030 under its Sustainable Africa Scenario (IEA, 2022). PV energy is expected to experience significant growth due to its competitiveness and low-carbon nature. However, solar production is highly dependent on weather conditions (Dajuma et al., 2016).

The growth of solar energy in West Africa calls for the development of tailored tools to facilitate its integration into power grids and ensure optimal operational maintenance. Accurate production forecasts are required by solar power plant operators, spanning various timescales, ranging from a few hours to several days. This is essential for maximising production, reducing penalties linked to predicted deliverable energy, and optimising plant maintenance to minimise production losses. High-quality forecasts are also crucial for electricity grid operators to maintain supply-demand equilibrium and ensure system stability. Therefore, the variability of energy production significantly affects them. The key

meteorological variables that influence photovoltaic production are the Global Horizontal Irradiance (GHI) and the air temperature. These factors, which directly impact electricity production and cell efficiency, often reach high levels in this region as demonstrated by Dajuma et al. (2016) and Ziane et al. (2021). Their findings indicate that solar irradiance is the primary factor influencing PV production, as the generated current by the photoelectric effect is proportional to the irradiance. Furthermore, they demonstrate that, at the second order, the air temperature affects the efficiency of solar cells, as both parameters are inversely correlated.

Clauzel et al. (2024) identified desert dust aerosol as a significant source of GHI forecast errors for the only two solar power plants in the Sahel region of Sococim (Senegal) and Zagtouli (Burkina Faso), particularly during the dry season. Dust aerosols are a key element in the West African climate and strongly influence solar farm production through their direct effect (aerosol-radiation interaction (ARI), Briant et al., 2017) and indirect effects (aerosol-cloud interaction (ACI), Tuccella et al., 2019) on radiation, and also through their deposition on solar panels (fouling effect, Diop et al., 2020, Aidara et al., 2023). As mentioned by Kok et al. (2021), the West African desert aerosol load is the highest in the world and occurs mainly during the dry season. In fact, North Africa, including the Sahara, is the world's largest contributor to desert dust emissions (Prospero et al., 2002), and 60% of this dust is transported to the West African region (D'Almeida, 1986; Kok et al., 2021). Most dust emissions are associated with synoptic-scale atmospheric dynamics such as the Harmattan flow during the dry season (Klose et al., 2010). Engelstaedter and Washington (2007) pointed out the importance of small-scale wind events associated with the large-scale flow, especially in the Bodele depression, which is a hotspot for dust emissions (Engelstaedter et al., 2006). Analysing satellite observations, Schepanski et al. (2009) show that 65% of the activation of the dust source area occurred in the early morning, demonstrating the important role of the breakdown of the nocturnal low-level jet. Washington and Todd (2005) confirmed the importance of the Bodele low-level jets during the dry season in initiating dust emissions that can be transported to the West African coast within a few days. Dust aerosol emissions are also highly linked to Mesoscale Convective Systems (MCS, Marsham et al., 2008 ; Bergametti et al., 2017) and to strong near-surface winds in the intertropical discontinuity zone during the rainy season (Bou Karam et al., 2009).

Some studies intend to model dust events in West Africa such as Ochiegbu (2021) who implemented a back-trajectories model to understand the dust event reaching Nigeria. This work revealed that most of the aerosols coming to Nigeria between 2011 and 2014 were originating from the Bodele Depression. Menut (2023) focused on dust forecasting during the Cloud-Atmospheric Dynamics-Dust Interactions in West Africa (CADDIWA) campaign during summer 2021 (Flamant et al., 2024) using the CHIMERE regional chemistry-transport model (Menut et al., 2021). The model was coupled online with the Weather Research and Forecasting (WRF) meteorological model (Briant et al., 2017; Tuccella et al., 2019) to perform dust aerosol concentration forecasts. The results of this work provide confidence in the model coupling in the region as the dust forecast quality does not decrease with time over a few days. In addition, only a limited number of studies have been conducted on the prediction of GHI in the West African region. Sawadogo et al. (2024) conducted an evaluation of WRF-solar GHI forecast (Jimenez et al., 2016) in Ghana for the year 2021. In their work, a version of the model coupled offline with Copernicus Atmosphere Monitoring Service (CAMS) Aerosol Optical Depth (AOD) forecasts was considered to integrate

information on aerosol load. They showed that WRF-Solar outperforms in predicting GHI under clear sky conditions while its performance under high aerosol levels remains poor, that was mainly attributed to uncertainties in the input AOD during data assimilation within the model. Close to the region of interest, for the northern Morocco area, El Alani et al. (2020) compared the performance of global models (Global Forecast System, Integrated Forecast System, McClear) and demonstrated their proficiency in capturing GHI hourly temporal variability.

As far as our knowledge is concerned, no studies have been conducted to assess online coupled simulations between a meteorological model and an aerosol life cycle model representing the emissions, the transport and the deposition in West Africa to estimate solar irradiance. This is despite the significant presence of desert dust, characterised by high concentrations in the region. Additionally, scarce attention has been given to the significance of initial and boundary conditions for conducting the aerosol model on the performance of analysis simulations, and to our knowledge, investigating these aspects would represent a novel contribution to research in the West African region.

Within this general context, the objectives of this study are two folds i) to evaluate the ability to reproduce a dust event using a meteorological and dust life cycle model coupling configuration, and ii) to investigate whether the performance of the simulations can be enhanced by modifying the aerosol initial and boundary conditions employed, and to estimate the uncertainty associated with this dataset selection with regard to the errors made by the model. Section 2 introduces the case study, the simulation configuration, the data and models selected for this work. In Section 3, the results are presented, beginning with the variables of interest for solar production (GHI and surface air temperature), followed by the variables associated with the desert aerosols (AOD, concentration, size distribution, emissions). Section 4 gives main conclusions and draws some perspectives for this study.

## 2. Material and methods
### 2.1. Case study
The case study is a dust event that occurs in West Africa from March 26th-00 UTC to April 2nd-00 UTC, 2021, i.e., during the dry season. High dust emissions occur at the Bodélé Depression (Chad), the plume being then transported westward. The dust plume reached its maximum intensity in terms of AOD and dust concentration over West Africa, and in particular over the Zagtouli solar farm (Burkina-Faso, Fig. S1), on March 30th. The event was also chosen because it was not predicted in the solar forecast currently implemented for the Zagtouli solar farm, leading to solar forecast errors during the passage of the dust plume (Clauzel et al., 2024).

Figure 1 illustrates that this event is characterised by a strong Harmattan flow, with surface winds from the South/South-West sweeping across the Bodélé Depression (Chad), where the potential for desert dust emissions is very high (Prospero et al., 2002; Washington et al., 2006). Additionally, this event is characterised by a westward flow between Chad and the Atlantic coast, which facilitates the transportation of the dust plume. Fig. 1a shows MODerate-resolution Imaging Spectroradiometer (MODIS) satellite observations of the AOD, identifying the initial dust source area on the Bodélé Depression, as well as the westward movement of the plume. This event is characteristic of the West African dry season climatology, with a dominant Harmattan flow as described in the introduction. Figure S1 provides further insight into the dust plume transport during the case study.

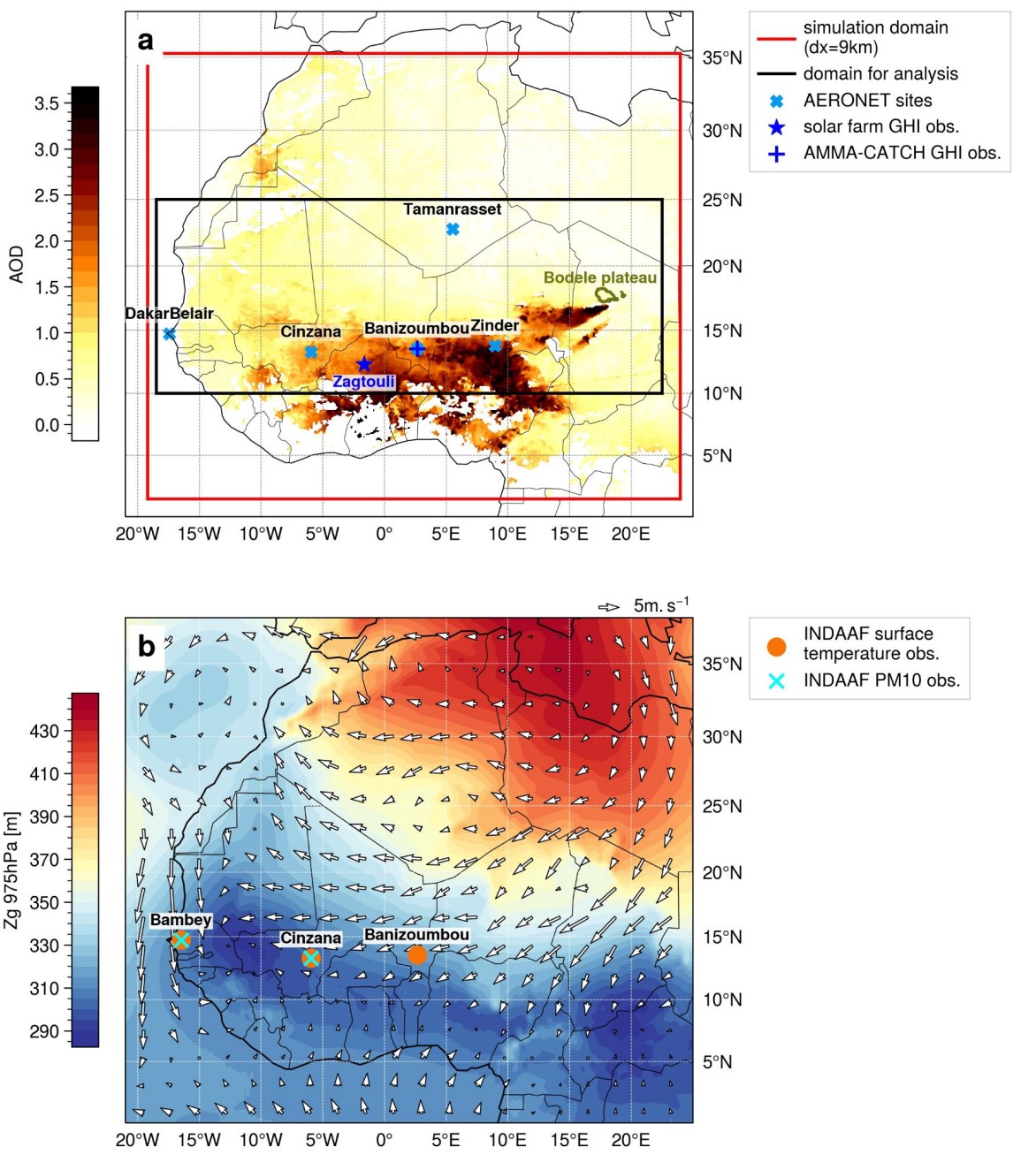

**Figure 1 -** a) Mean aerosol Optical Depth at 550nm from MODIS satellite observations over the period 28 March-00 UTC to 02 April-00 UTC 2021. The Global Horizontal Irradiance (GHI) observations and AERONET aerosol measurement network, introduced in 2.4, are presented, as well as the boundaries of the simulated domain (red rectangle) and the area of interest for analysis (black rectangle). b) Mean synoptic conditions of the geopotential height (Zg) at 975hPa and the 10m-wind (white arrows - in m/s) over the period 28 March-00 UTC to 02 April-00 UTC 2021 from ERA5 reanalysis. The surface temperature and aerosol concentration observations from the INDAAF network, introduced in 2.4, are presented.

## 2.2. Modelling tools
### 2.2.1. WRF model

The meteorological Weather and Research and Forecasting model (WRF) model version
3.7.1 is taken for compatibility with the CHIMERE coupling procedure. It is used in its non-
hydrostatic configuration (Skamarock et al., 2008) and is forced at the boundaries of the
domain every hour by the meteorological reanalysis data of ERA5 (ECMWF) provided on a
regular 0.25° x 0.25° grid.
The model is run with a 9 km horizontal resolution, a 45s integration time step and 50
vertical levels, from the surface to 50 hPa. The updated Rapid Radiative Transfer Model
(RRTMG) radiation scheme (Iacono et al., 2008), which is mandatory for the aerosol optical
properties feedback, is employed for both long- and short-wave radiations. Additionally, the
Thompson aerosol-aware microphysics scheme (Thompson and Eidhammer, 2014) is
applied. The Yonsei University planetary boundary layer's surface layer scheme (Hu et al.,
2013) is also used, and the cumulus parameterisation is based on the Grell-Freitas scheme
(Arakawa, 2004). The Revised MM5 surface layer scheme (Jiménez et al., 2012) is
employed, while the Noah-MP Land Surface Model (Niu et al. 2011) is implemented for the
land surface physics scheme.
### 2.2.2.  CHIMERE model

The chemistry-transport model CHIMERE version v2020r3 (Menut et al., 2021) is used in
conjunction with the WRF model. Both models have a 9 km horizontal grid. The CHIMERE
model has 30 pressure-dependent vertical levels from the surface up to 200 hPa, with a first
layer thickness of 3 hPa. The model is configured for dust-only, with no chemistry and only
considering dust aerosols (details in section 2.3). The threshold friction velocities for dust
emission are estimated using the Shao and Lu scheme (2000) and the 6-km spatial
resolution GARLAP (Global Aeolian Roughness Lengths from ASCAT and PARASOL)
dataset from Prigent et al. (2012). Mineral dust emission fluxes were calculated employing
the Alfaro and Gomes (2001) scheme on 10 aerosol size bins ranging from 0.01 to 40 µm.
The Fécan et al. (1999) parametrization is employed to account for the inhibitory effect of
soil moisture on dust emission. Dry deposition is treated as described in Zhang et al. (2001).
Wet scavenging for aerosol is computed following the Willis and Tattelman scheme (1989).
The CHIMERE model includes the Fast-JX module, version 7.0b (Wild et al., 2000; Bian et
al., 2002) for the calculation of radiative processes. It considers the radiative properties for
each aerosol species and each aerosol size bin independently to compute the aerosol
optical depths, the single scattering albedo and the aerosol asymmetry factor. More details
on the dust aerosol radiative properties are given in Tables S1 and S2. Finally, we test three
different initial and boundary condition datasets for mineral dust load (see 2.2.3).
**Table 1 -** Parameterizations used in WRF and CHIMERE

| WRF | |
| --- | --- |
| microphysics | Thompson aerosol-aware (Thompson and Eidhammer, 2014) |
| radiation | RRTMG scheme for LW and SW (Iacono et al., 2008) |
| land surface | Noah-MP land surface scheme (Niu et al., 2011) |
| planetary boundary layer | Yonsei University scheme |

|  | (Hu et al., 2013) |
| surface layer | Revised MM5 surface layer scheme (Jimenez et al., 2012) |
| cumulus | Grell-Freitas scheme (Arakawa, 2004) |

| **CHIMERE** | |
| --- | --- |
| threshold friction velocities | Shao and Lu (2000) scheme |
| soil moisture | Fécan et al. (1999) scheme |
| dust emission fluxes | Alfaro and Gomes (2001) scheme |
| radiative processes | Fast-JX model, version 7.0b (Wild et al., 2000; Bian et al., 2002) |
| aerosol size distribution bins (diameters in μm) | 0.010 - 0.022<br>0.022 - 0.048<br>0.048 - 0.107<br>0.107 - 0.235<br>0.235 - 0.516<br>0.516 - 1.136<br>1.136 - 2.500<br>2.500 - 5.000<br>5.000 - 10.00<br>10.00 - 40.00 |


**2.2.3.**     **Dust aerosol initial and boundary condition datasets**

In this study, the uncertainty in the solar estimate associated with the initial and boundary
conditions of the dust aerosol load is evaluated. Three datasets were used: a climatology
derived from the Global Ozone Chemistry Aerosol Radiation and Transport (GOCART,
Ginoux et al., 2001), the Modern-Era Retrospective analysis for Research and Applications
Version 2 (MERRA2) reanalysis (Gelaro et al., 2017) and the CAMS reanalysis (Inness et
al., 2019).
The GOCART climatology is provided with the distribution of the CHIMERE model. It is a
monthly climatology on a coarse horizontal grid (2°x2.5°), which is corrected by applying a
factor of 0.3 as in Vautard et al. (2005).
The MERRA2 reanalysis combines the Goddard Earth Observing System (GEOS) and
GOCART models, which are online coupled and implemented with a data assimilation
system. It has a 3-hour temporal resolution and is presented on a 0.5°x0.635° horizontal
grid. The observational data considered in the data assimilation process are AOD satellite
observations from MODIS, Advanced Very High Resolution Spectroradiometer (AVHRR),
Multi-angle Imaging SpectroRadiometer (MISR) and ground observations from the AErosol
RObotic NETwork (AERONET).
The CAMS reanalysis was constructed using 4DVar data assimilation in ECMWF's
Integrated Forecast System (IFS). It has a temporal resolution of 3 hours and is computed
on a regular 0.75° horizontal grid. The AOD data from the Visible Infrared Imaging
Radiometer Suite (VIIRS), the MODIS and the Infrared Atmospheric Sounding Interferometer
(IASI) satellite observations are used as observational information in the data assimilation
process. The version 48R1 of CAMS is used in this study.
These three dust aerosol initial and boundary datasets differ in type (climatological or
reanalysis), in horizontal, vertical and temporal resolution, and in the resolution and range of
their aerosol size distribution. While GOCART has the highest number of aerosol classes
with 7 bins, CAMS covers a wider size spectrum despite a lower size resolution with only 3
classes. MERRA2 has an intermediate resolution with 5 classes, but covers a smaller
particle size spectrum than CAMS. The CHIMERE model pre-processes these dust aerosol
size distributions by applying a transfer coefficient $\delta$ to compute the dust aerosol
concentration on the 10 aerosol size bin defined for the simulations :

$$c_j = \sum_i \delta_{i,j} \times c_i$$

(1)

where $c_i$ is the dust aerosol concentration of the $i^{th}$ size bin from the initial and boundary
condition dataset considered, $c_j$ is the dust aerosol concentration of the $j^{th}$ size bin in the
CHIMERE simulation, and $\delta_{i,j}$ is the transfer coefficient. This transfer coefficient is derived
as :
-   $\delta_{i,j} = 0$ if the $i^{th}$ size bin from the initial and boundary condition dataset is found to be
wholly outside the $j^{th}$ size bin in the CHIMERE simulation;
-   $\delta_{i,j} = 1$ if the $i^{th}$ size bin from the initial and boundary condition dataset is wholly
encompassed by the $j^{th}$ size bin in the CHIMERE simulation;
-   $\delta_{i,j} = \dfrac{\log(r_{j,max}) - \log(r_{j,min})}{\log(R_{i,max}) - \log(R_{i,min})}$ if the $i^{th}$ size bin from the initial and boundary condition
dataset wholly encompasses the $j^{th}$ size bin in the CHIMERE simulation;
-   $\delta_{i,j} = \dfrac{\log(R_{i,max}) - \log(r_{j,min})}{\log(R_{i,max}) - \log(R_{i,min})}$ if the $i^{th}$ size bin from the initial and boundary condition
dataset partially overlaps the $j^{th}$ size bin in the CHIMERE simulation, but extends
below the start of this size bin;
-   $\delta_{i,j} = \dfrac{\log(r_{j,max}) - \log(R_{i,min})}{\log(R_{i,max}) - \log(R_{i,min})}$ if the $i^{th}$ size bin from the initial and boundary condition
dataset partially overlaps the $j^{th}$ size bin in the CHIMERE simulation, but extends
beyond the end of this size bin;
where $R_{i,min}$ and $R_{i,max}$ are respectively the radius of the lower and upper limit of the $i^{th}$ size
bin from the initial and boundary condition dataset, and $r_{j,min}$ and $r_{j,max}$ are respectively the
radius of the lower and upper limit of the $j^{th}$ size bin in the CHIMERE simulation.
For the sake of simplicity, throughout this article, we will refer to the WRF-CHIMERE
simulations runned with the GOCART, the MERRA2, and the CAMS dust aerosol initial and
boundary conditions as *wrf_chimere-G*, *wrf_chimere-M*, and *wrf_chimere-C* simulations
respectively.
Table 2 summarises the characteristics of the three dust aerosol datasets and their
associated size distributions.

**Table 2.** Summary of the characteristics of the dust initial and boundary condition products.

|  | **GOCART** | **MERRA2** | **CAMS** |
|---|---|---|---|
| type | climatology | reanalysis | reanalysis |
| temporal resolution | monthly | 3h | 3h |
| vertical levels | 20 | 72 | 60 |
| horizontal resolution (lat x lon) | 2°x2.5° | 0.5°x0.635° | 0.75°x0.75° |
| dust aerosol size distribution (radius in µm) | 0.20 - 0.36 µm | 0.1 - 1.0 µm | 0.03 - 0.55 µm |
|  | 0.36 - 0.60 µm | 1.0 - 1.8 µm | 0.55 - 0.90 µm |
|  | 0.60 - 1.20 µm | 1.8 - 3.0 µm | 0.90 - 20.00 µm |
|  | 1.20 - 2.00 µm | 3.0 - 6.0 µm | |
|  | 2.00 - 3.60 µm | 6.0 - 10.0 µm | |
|  | 3.60 - 6.00 µm | | |
|  | 6.00 - 12.00 µm | | |

### 2.3.    Modelling strategy
The domain of simulation extends from 2° to 35°N and from 19°W to 24°E, , as illustrated by
the red box in Figure 1b. The domain is large enough to represent the primary atmospheric
flows, including the Harmattan North/North-West flow and the monsoon South flow, as well
as the transport of the emitted aerosol plumes. A horizontal resolution of 9 km has been
selected in order to ensure that the grid ratio is approximately 3 with the ERA5
meteorological forcing. This choice is also motivated by the a priori intention to achieve a
resolution higher than that of previous CHIMERE simulations performed in this region and
compared to the operational solar forecast model used for the Zagtouli solar farm, which are
based on global forecast models (see 2.4.1). The CHIMERE model is configured in a "dust
only" model, which models only the mineral dust type. This hypothesis is supported for this
dust case study by Fig. S2, as desert dust is the dominant aerosol during the event,
particularly above 10°N. This hypothesis is also reinforced by the dust optical depth (DOD)
to AOD ratio derived from the CAMS reanalysis, which exceeds 80% during this case study
and for the domain of interest (not shown). It is notable that biomass burning, which
represents the other principal aerosol source in this region, is no longer a significant
contributor to aerosol levels at that time of the year (Evans et al., 2018).
The WRF and CHIMERE models are coupled online through the OASIS3 MCT coupler. A
two-way coupling strategy is selected, in which WRF sends meteorological variables to
CHIMERE which in turn exchanges aerosol information such as AOD, Single Scattering
Albedo (SSA) and Asymmetry Factor. This coupling strategy imposes most of the WRF
parameterisations. The exchange frequency is set to 15 minutes. The WRF model computes
fields on 50 levels, which are linearly interpolated over the 30 CHIMERE vertical levels via
the OASIS coupler. The coupling includes the feedbacks of aerosol-radiation interactions
(ARI, direct aerosol effect) and aerosol-cloud interactions (ACI, indirect aerosol effects)
simultaneously.
The simulation starts on March 14th-00 UTC and ends on April 2nd-00 UTC, 2021. The first
two weeks served as the spin-up period. The simulation outputs are analysed for the period
of March 28th-00 UTC UTC to April 2nd-00 UTC, which corresponds to the passage of the
dust plume in the Sahel region, in particular around the Zagtouli solar farm in Burkina Faso.
Four simulations were conducted: a meteorological simulation using WRF model alone, and
dust simulations with the coupled WRF-CHIMERE models using as initial and boundary
conditions the GOCART climatology, the MERRA2 reanalysis and the CAMS reanalysis. The
simulation using only WRF allows for the evaluation of the impact of taking into account dust
aerosols in estimating solar irradiance. This is compared to the other three simulations,
which are also used to evaluate the uncertainties associated with the choice of the aerosol
initial and boundary condition dataset. A domain of interest, spanning 10°N to 25°N (Fig. 1a),
was selected for analysis and comparisons. This choice was guided by the dust plume
trajectory (Fig. S1) and the "dust only" hypothesis (Fig. S2).

### 2.4.    Evaluation datasets

This section presents the local and regional data that are employed in the evaluation of the
simulations.

#### 2.4.1.    GHI

The Global Horizontal Irradiance (GHI) is the total shortwave irradiance from the Sun on a
horizontal surface on Earth.  It is the sum of direct irradiance, which takes into account the
solar zenith angle, and diffuse horizontal irradiance. It is measured in   $W.m^{-2}$ for the
wavelength range 0.3 - 3.0 μm.

The national electricity company of Burkina-Faso, Sonabel, operates a solar farm in Zagtouli
(12.31°N;1.64°W; Fig. 1a), approximately 15 km west of the capital, Ouagadougou. It has an
installed capacity of 34 MWp and contributes up to 4% of Burkina Faso's annual electricity
production. Ground GHI measurements from pyranometers are available at a temporal
resolution of 15 minutes for the Zagtouli solar plant and undergo pre-processing to ensure
quality control. This involves removing outliers and days with missing data, visually checking
the consistency of the measured values and selecting data corresponding to production
hours (positive values for solar irradiance at the top of the atmosphere). Operational GHI
forecasts for this solar farm are computed by the French company Steadysun. These
forecasts are based on a multi-model, multi-member and multi-mesh grid aggregation, which
is derived from the NCEP Global Ensemble Forecast System and the ECMWF Integrated
Forecast System (Clauzel et al., 2024).
In-situ measurements of GHI from pyranometers (Fig. 1a) are also available at a 15-minutes
temporal resolution for the Banizoumbou (Niger) surface station, installed as part of the
AMMA-CATCH observatory (Analyse Multidisciplinaire de la Mousson Africaine - Couplage
de l'Atmosphère Tropicale et du Cycle Hydrologique, AMMA-CATCH (2005)).
The two measurement sites were selected because they are the only locations where GHI
observations have been made available along the dust plume transport for the case study,
with the Zagtouli power station being one of the first large solar farms in West Africa and the
AMMA-CATCH observatory being the only one to offer continuous GHI measurements for
the region and period of interest.

The CAMS gridded solar radiation dataset (CAMS solar radiation services v4.6, Schroedter-Homscheidt et al., 2022), based on the Heliosat-4 method (Qu et al., 2017), provides several variables related to solar irradiance, such as clear-sky and all-sky GHI. It has a horizontal resolution of 0.1°x0.1° and provides data every 15 minutes. The clear sky model includes aerosols through the CAMS chemical transport model (Inness et al., 2019), which integrates data assimilation of AOD and is coupled online to a numerical weather prediction model. Cloud information for the all-sky model is derived from MeteoSat Second Generation (MSG) satellite observations using the AVHRR Processing scheme Over cLouds, Land and Ocean (APOLLO) Next Generation cloud processing scheme (Klüser et al., 2015). The dataset was selected for comparison with the simulations as it integrates a description of aerosol processes. While Yang and Bright (2020) and Sawadogo et al. (2023) show that it is the best performing product for estimating surface solar irradiance in the West African region among several satellite-based gridded irradiance products, this dataset still has a negative bias of about 10% for all-sky solar irradiance estimates at desert stations in North Africa (CAMS solar radiation regular validation report, Lefèvre, 2022).

### 2.4.2. Surface temperature

In-situ surface temperature measurements are available for three stations of the International Network to study Deposition and Atmospheric composition in Africa (INDAAF) : Banizoumbou (Niger, 13.54° N, 2.66° E, 6.2m above surface; Rajot et al, 2010a; Marticorena et al, 2010; Kaly et al., 2015), Cinzana (Mali, 13.28° N, 5.93° W, 2m above surface; Rajot et al, 2010b; Marticorena et al, 2010; Kaly et al., 2015) and Bambey (Senegal, 14.70° N, 16.47° W, 5.2m above surface; Marticorena et al, 2021a) (Fig. 1b). The measurement sites were selected since they are almost aligned around 13-15° North, which represents the main pathway of Saharan and Sahelian dust towards the Atlantic Ocean during the case study.
The ERA5 atmospheric reanalysis (Hersbach et al., 2020) provides spatially continuous hourly values of surface temperature at 2 metres and has a horizontal resolution of 0.25° x 0.25°.

### 2.4.3. Aerosol

The INDAAF network also provides data on aerosol concentration through ground measurements of $PM_{10}$, i.e. the concentration of atmospheric particles having an aerodynamic diameter less than 10 µm. For this case study, hourly $PM_{10}$ measurements are available for two stations (Fig. 1b): Cinzana (Rajot et al, 2010c; Marticorena et al, 2021; Kaly et al, 2015) and Bambey (Marticorena et al, 2021b).
The CAMS atmospheric reanalysis (Inness et al., 2019) is also used to evaluate regional surface $PM_{10}$ concentration and AOD. It provides 3-hourly data with a horizontal resolution of 0.75° x 0.75°, with a surface layer thickness of 2.4 hPa.

Local ground measurements of AOD are retrieved from the AErosol RObotic NETwork level 1.5 dataset (AERONET, Holben et al., 1998; Giles et al., 2019). AOD is calculated from sun photometer recordings, along with Ångström Exponent, and is only available during clear sky conditions in daylight hours, with a resolution of 1 minute. The AOD at 400 nm simulated with the WRF-CHIMERE model is converted to 440 nm for comparison with AERONET, using the Ångström formula :

$$\frac{AOD_\lambda}{AOD_{\lambda_0}} = \left( \frac{\lambda}{\lambda_0} \right)^{-\alpha} \tag{2}$$

where $AOD_\lambda$ is the AOD at the desired wavelength, $\lambda = 440\,nm$ here ; $AOD_{\lambda_0}$ is the AOD at
the wavelength simulated in the model, $\lambda_0 = 400\,nm$ here ; $\alpha$ is the Ångström exponent,
derived from the simulated AOD at different wavelengths and here given for the range from
400 nm to 600 nm.
AERONET also provides an aerosol size distribution dataset estimated through inversion of
the photometers data, as described in Dubovik and King (2000). The algorithm for inversion
provides a volume particle size distribution for 22 bins, which are logarithmically distributed
for radii between 0.05 µm and 15 µm. For comparison with the modelled aerosol size
distribution, this distribution is interpolated on the CHIMERE simulated aerosol size
distribution which is composed of 10 bins ranging from 0.01 µm to 40.00 µm in diameter (see
Table 1). Given that the coarsest bin (10.00-40.00 µm) is at the limit of the capabilities of the
inversion method, and the two thinnest bins (0.010-0.022 µm and 0.022-0.048 µm) are out of
the range of the inversion product, the AERONET dataset size sections are interpolated on
the CHIMERE size sections ranging from 0.048 to 10.0 µm. Consequently, only comparisons
between the three simulations can be made for the three size sections which are out of the
range of AERONET product. The column aerosol volume size distribution simulated by the
model is calculated for each bin "i" as in Menut et al. (2016) :

$$\frac{dV(r_i)}{d\ln(r_i)} = \sum_{k=1}^{nlevels} \frac{m_{k,r_i} \times \Delta z_k}{\rho_{dust} \times \ln(r_{i,max}/r_{i,min})} \tag{3}$$

where $r_i$ is the mean mass median radius (in µm) and $r_{i,min}$ and $r_{i,max}$ the boundaries of the
$i^{th}$ bin. $m_{k,r_i}$ is the dust aerosol mass concentration (the mass of aerosol in one cubic metre
of air, in $\mu g.m^{-3}$). $\rho_{dust}$ is the dust aerosol density (the mass of the particle in its own volume,
$\rho_{dust} = 2300\,kg.m^{-3}$). $\Delta z_k$ is the model layer thickness (in metres), for a total of n levels (here
30 vertical levels).
The locations of the five AERONET sites used for comparison in this study are illustrated in
Figure 1a.
The spatially continuous AOD is also derived from level 2 aerosol products of MODIS Terra
and Aqua satellites (combined Dark Target, Deep Blue AOD at 0.55 micron, Collection 6.1,
Platnick et al., 2015). It provides a measure of the AOD at 550 nm during daytime for clear
sky conditions, with a spatial resolution of 10 km. To compare simulated AOD from WRF-
CHIMERE models with AOD from MODIS, the former is converted from 600 nm to 550 nm.
The conversion is performed using the Ångström formula (eq. 2).
Table 3 provides a general overview of the data used to evaluate the simulations in this
study.
**Table 3 -** Summary of data used to evaluate the simulations.

| | product | type | resolution |
|---|---|---|---|
| | Zagtouli solar farm monitoring system | pyranometer GHI measurement | local |

| | | | |
|---|---|---|---|
| **GHI** | AMMA-CATCH observational network | pyranometer GHI measurement | local |
| | CAMS gridded solar radiation | atmospheric reanalysis | 0.01°x0.01° |
| **temperature** | INDAAF network | ground measurements | local |
| | ERA5 | atmospheric reanalysis | 0.25°x0.25° |
| **PM$_{10}$** | INDAAF network | ground measurements | local |
| | CAMS (v48R1, EAC4) | atmospheric reanalysis | 0.75°x0.75° |
| **Aerosol Size Distribution** | AERONET network | inversion product | local |
| **Aerosol Optical Depth** | AERONET network | sunphotometer ground measurements | local |
| | MODIS | satellite observations | 10km |


## 3. Results

The analysis starts by assessing the errors and uncertainties associated with the dust
aerosol initial and boundary condition dataset employed to estimate the variables of interest
for solar production, i.e. GHI and surface temperature. Subsequently, we investigate the
potential causes of these uncertainties by evaluating the AOD, aerosol size distribution, and
surface aerosol concentration (PM$_{10}$), as well as by examining mineral dust emissions and
the flux of these aerosols at the boundaries of the domain. The metrics used to assess the
quality of the simulations are described in Supplementary Materials.

### 3.1. GHI

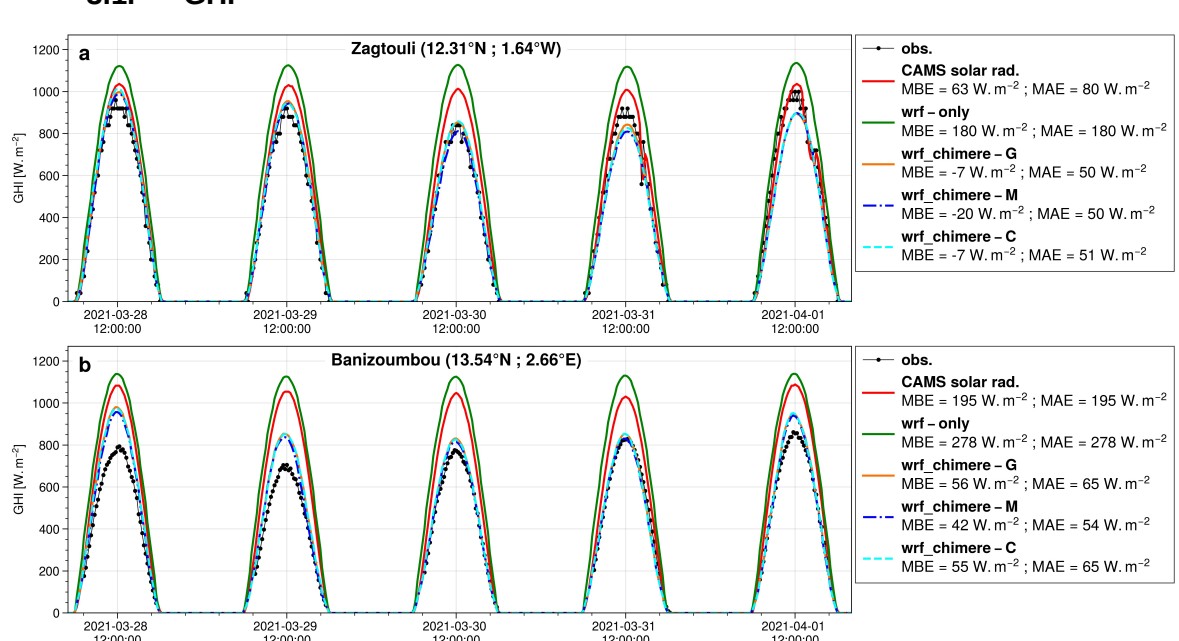


**Figure 2 -** Local comparison of CAMS gridded solar radiation product and simulated GHI against a) the Zagtouli solar farm observations and b) the Banizoumbou AMMA-CATCH observations. *wrf_chimere-G*, *wrf_chimere-M* and *wrf_chimere-C* refer to the WRF-CHIMERE simulations using GOCART, MERRA2 and CAMS as dust aerosol initial and boundary condition dataset respectively.

In Fig. 2, the local evaluation demonstrates the effect of taking into account dust aerosol for GHI estimation with the WRF-CHIMERE coupling over the WRF meteorological model alone. The coupling reduces the MAE by a factor of 3.6 at Zagtouli and by a factor of 4.6 at Banizoumbou on average. The simulations accurately represent the reduction in GHI intensity caused by the dust plume at both stations. However, the reduction persists compared to the observations at Zagtouli. At Banizoumbou, the simulations overestimate GHI at the beginning and end of the case study.

Figure 2 also indicates that the CAMS gridded solar radiation product fails to fully reproduce the dust event, with only a small reduction in GHI during the passage of the dust plume and an intermediate MAE between the WRF only and the WRF-CHIMERE simulations. This point serves to highlight the advantages of using a regional model in comparison to a global product for the simulation of dust conditions and the estimation of solar irradiance.

Furthermore, the uncertainty in GHI estimation related to the choice of the dust aerosol initial and boundary condition dataset is limited, particularly when compared to the errors. This is evidenced by the fact that the mean standard deviation between the three WRF simulations is only 7% of the average MAE of these simulations at Zagtouli, and only 5% at Banizoumbou.

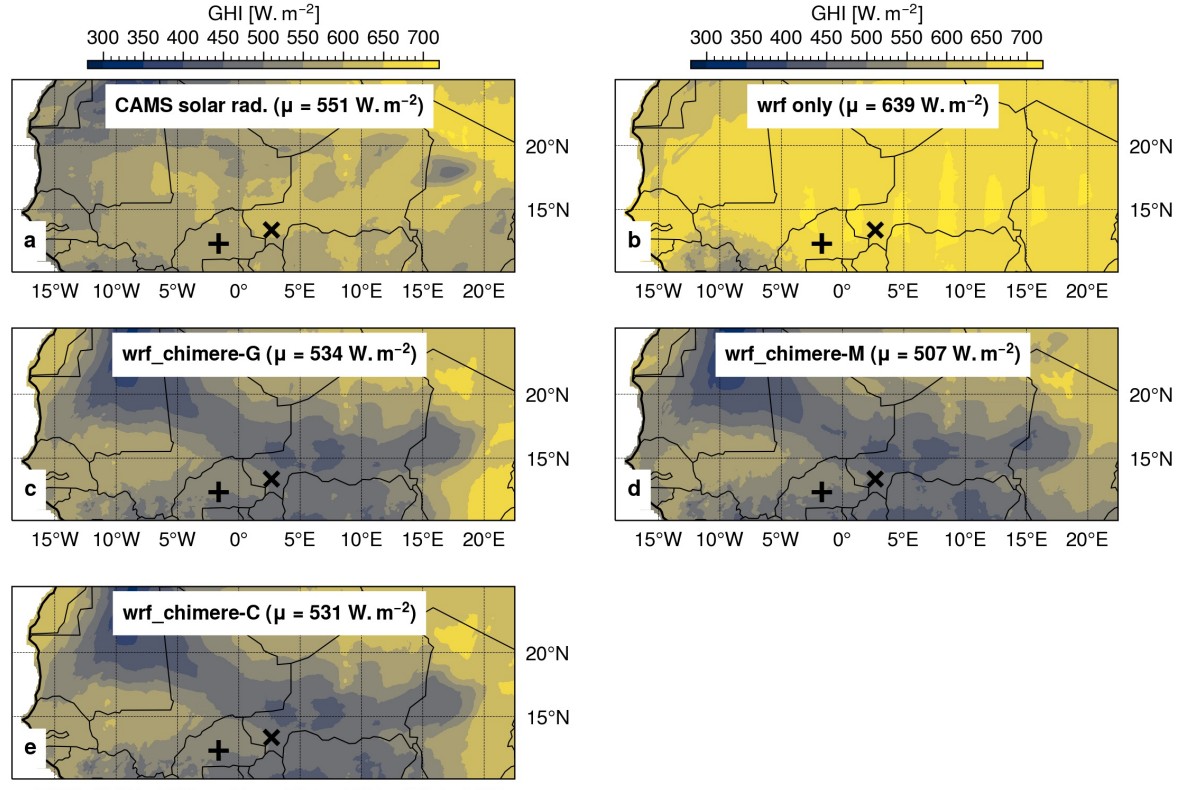

**Figure 3 -** Mean day-time GHI during the period of 28 March-00 UTC to 02 April-00 UTC 2021 as estimated by a) the CAMS gridded solar radiation dataset, b) the WRF only

simulation, and the WRF-CHIMERE simulations with c) GOCART, d) MERRA2 and e)
CAMS as dust aerosol initial and boundary condition dataset; **+** is the Zagtouli solar farm
and **x** is the Banizoumbou site. $\mu$ is the mean GHI estimates over the domain.
The regional comparison presented in Fig. 3 provides more insight into the impact of
incorporating dust on GHI estimation with the WRF-CHIMERE coupling, when compared to
the WRF meteorological model alone. As anticipated the WRF-only simulation has the
highest GHI estimates. The WRF-CHIMERE simulations indicate that dust aerosols reduce
the mean GHI estimation by approximately $115\,W.m^{-2}$ (-18%) as compared to the WRF-only
simulation, while the CAMS gridded solar radiation global product shows a reduction of
$88\,W.m^{-2}$ (-14%). The three WRF-CHIMERE simulations exhibit identical regional patterns,
with lower mean GHI values observed on the dust plume trajectory from the Bodélé
Depression to the West, and also in the South Atlas region. In contrast, the CAMS gridded
solar radiation dataset does not show this regional pattern, which may indicate that this
global product does not fully capture the dust event.
Furthermore, the uncertainty in GHI estimation associated with the choice of the dust aerosol
initial and boundary conditions dataset is limited, particularly when compared to the changes
brought by the taking of dust aerosol into account. Indeed, the standard deviation between
the three WRF-CHIMERE simulations represents only 5% of the mean difference between
these three simulations and the WRF-only simulation without dust.
**3.2.   Temperature**

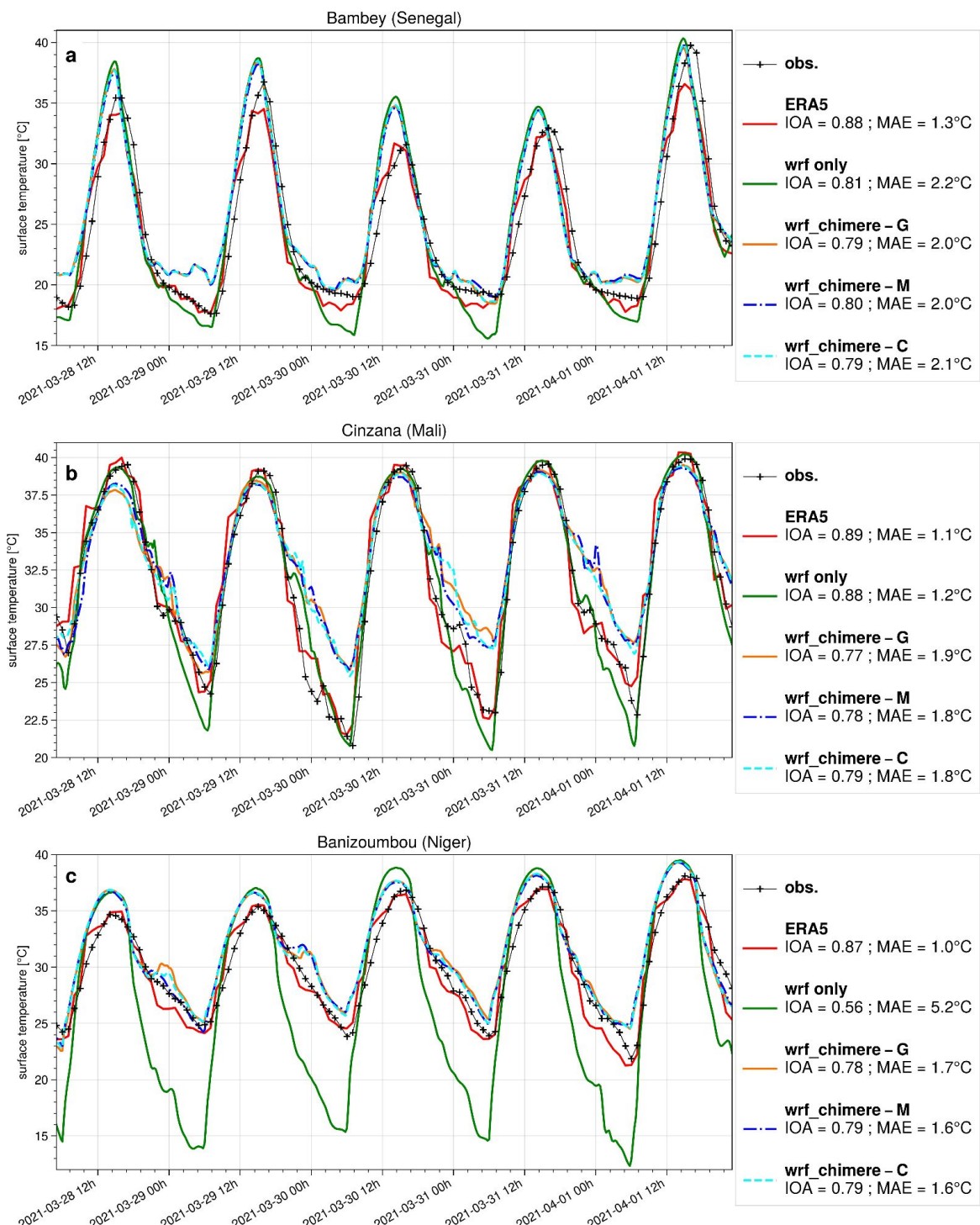


**Figure 4 -** Local comparison of ERA5 and simulated surface temperature with the INDAAF
observations for a) Bambey (Senegal), b) Cinzana (Mali) and c) Banizoumbou (Niger)
measurement sites. *wrf_chimere-G*, *wrf_chimere-M* and *wrf_chimere-C* refer to the WRF-
CHIMERE simulations using GOCART, MERRA2 and CAMS as dust aerosol initial and
boundary condition dataset respectively. *IOA* is the Indicator of Agreement and *MAE* is the
Mean Absolute Error.


Figure 4 illustrates the contrasting outcomes of taking into account dust aerosols into the
WRF-CHIMERE coupling in comparison to the WRF meteorological model alone for the

estimation of surface temperature. At Bambey (Fig. 4a), which is far from the dust source
areas, the coupling has no effect on daytime temperatures but does affect night-time
temperatures. The WRF-CHIMERE and WRF-only simulations have IOA and MAE of the
same order of magnitude. At Cinzana (Fig. 4b), the WRF-only simulation performed better,
with a MAE 0.6°C lower than the coupled simulations, especially for night-time temperatures
but also for estimating the daily temperature peak. Finally, at Banizoumbou (Fig. 4c), which
is near the dust source areas, the coupling leads to a significant improvement in surface
temperature estimation, with an IOA of approximately 0.79 compared to 0.56 for the WRF-
only simulation and a MAE reduced by around 3.6°C. The impact of dust aerosols on
temperature is particularly pronounced at night-time. However, dust also affects the daily
temperature peak, with a reduction of 1.1°C of the daily maximum temperature observed on
the 30th of March.
Depending on the position of the measurement station, the results show a contrast, with a
significant improvement with the model coupling close to the source zones at Banizoumbou.
However, this improvement is reversed with increasing distance at Cinzana. This suggests
errors in the simulation of the transport of the dust plume from the source zones (Bodélé
Depression) towards the West. Overall, the main differences between WRF only and WRF-
CHIMERE coupled simulations occur at night time when there is no solar production. These
differences highlight the warming effect due to the dust aerosol interaction with the longwave
earth radiation.
In general, the uncertainty associated with the choice of the dust aerosol initial and boundary
condition dataset for the WRF-CHIMERE simulations is negligible compared to the errors in
temperature estimation or the difference with the WRF-only simulation.
The value of the ERA5 reanalysis for surface temperature evaluation is also reinforced in
Fig. 4, since it shows the lowest MAE and highest IOA. This dataset can therefore be
considered reliable for a regional evaluation of surface temperature.

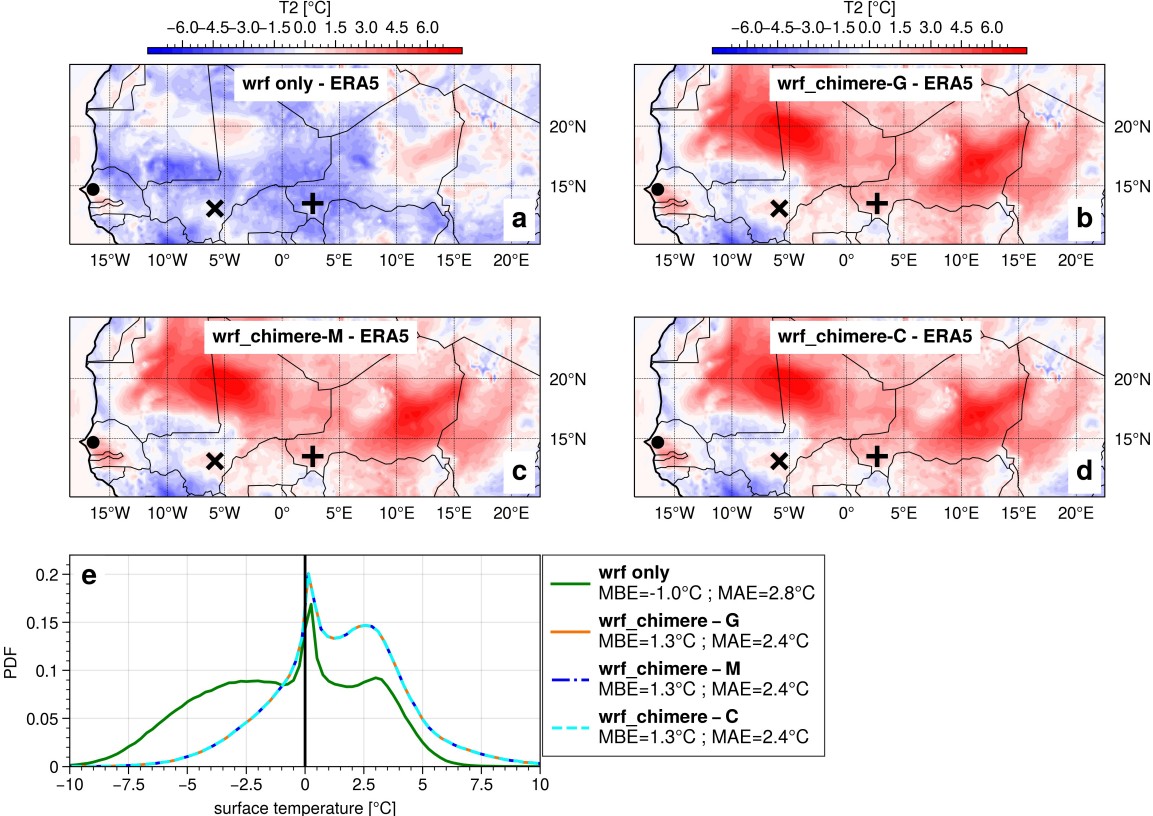

**Figure 5 -** Mean difference in surface temperature as compared to the ERA5 reanalysis for a) the WRF only simulation, the WRF-CHIMERE simulations with b) GOCART, c) MERRA2 and d) CAMS as dust aerosol initial and boundary condition dataset, during the period of 28 March-00 UTC to 02 April-00 UTC 2021; the black point is the Bambey, **x** is the Cinzana and **+** is the Banizoumbou INDAAF sites.  e) Probability Density Function for the differences in surface temperature between simulations and the ERA5 reanalysis.

The regional surface temperature evaluation in Fig. 5 also reveals a contrast benefit of the coupling approach for the surface temperature estimation. While the WRF alone simulation (Fig. 5a) underestimates the surface temperature all over the domain, WRF-CHIMERE simulations are overestimating surface temperature in the dusty areas (Saharan region, Fig. 5bcd). Overall, taking into account dust aerosol in the estimation of surface temperature reduces the MAE by 14% (Fig. 5e) when comparing the surface temperature estimates from simulations with the ERA5 reanalysis.

Furthermore, the uncertainty associated with the choice of the dust aerosol initial and boundary conditions dataset is limited. This is demonstrated by the fact that the standard deviation between the three WRF-CHIMERE simulations averaged over the period of analysis is 12% of the mean bias of those three simulations in comparison to ERA5 reanalysis, and only 7% of the difference between the coupled simulations and the WRF-only simulation without dust.

Finally, the incorporation of dust aerosol into the estimation of GHI appears to be a crucial element in this case study. However, the value of this approach is more debatable in the context of surface temperature estimation. Furthermore, the uncertainty related to the dust aerosol initial and boundary condition dataset selection is limited, particularly when

compared to the simulation errors, and to the differences between including dust in the
simulation and not including it. The following sections will examine the simulated dust
aerosol condition during the case study in order to explain the discrepancies observed in
GHI and surface temperature, which are key parameters for solar production.
### 3.3. Aerosol Optical Depth

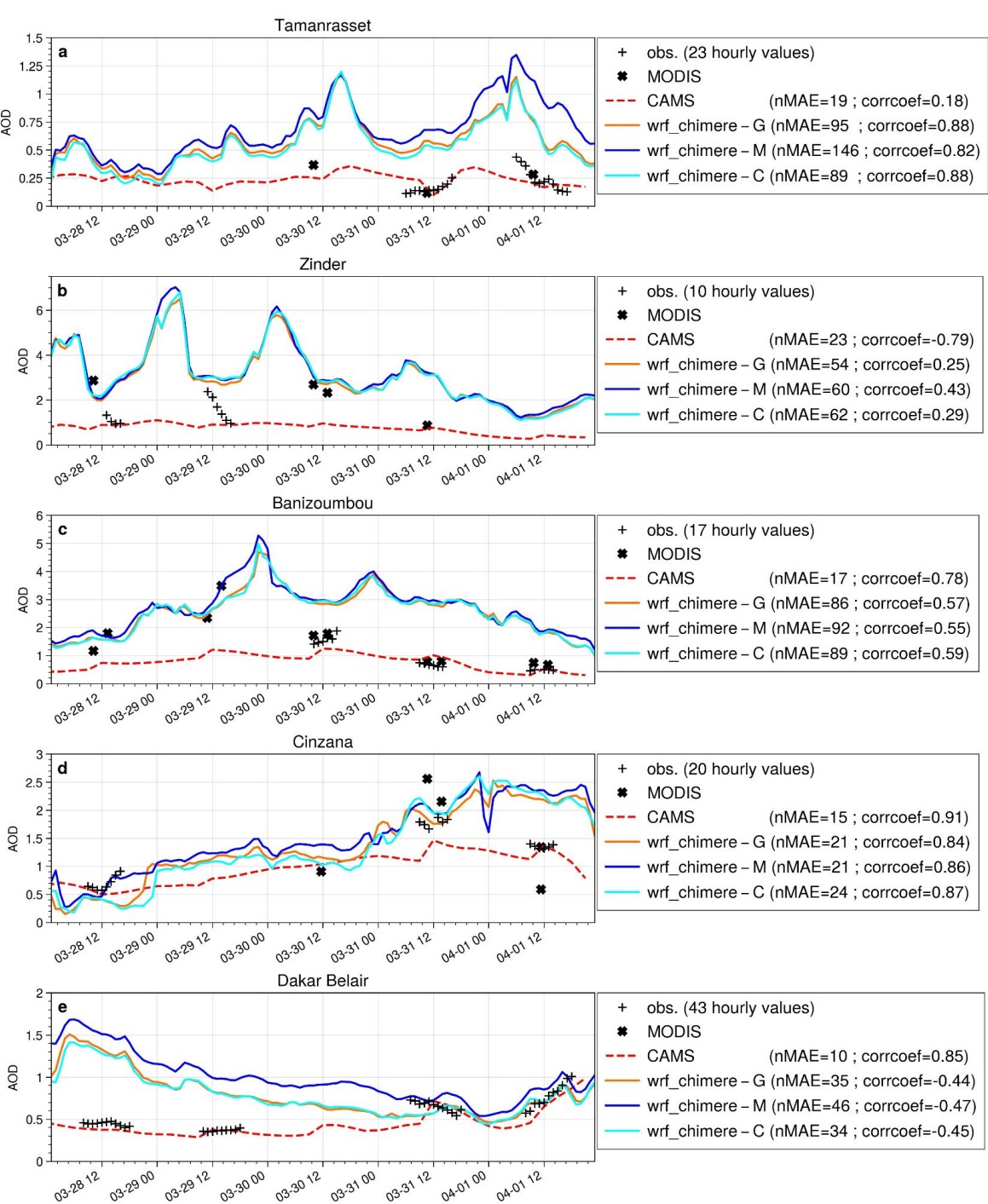

**Figure 6 -** Local comparison of simulated AOD with AERONET in-situ measurements at 440
nm for a) Tamanrasset, b) Zinder, c) Banizoumbou, d) Cinzana and e) Dakar Belair stations.
*wrf_chimere-G*, *wrf_chimere-M* and *wrf_chimere-C* refer to the WRF-CHIMERE simulations
using GOCART, MERRA2 and CAMS as dust aerosol initial and boundary condition dataset
respectively; *MODIS* and *CAMS* refer to the AOD at 440 nm from the MODIS satellite
observations and the CAMS atmospheric reanalysis respectively. $nMAE$ is the normalised
mean absolute error in % and $corrcoef$ is the Person correlation coefficient, both derived
with AERONET measurements as the reference.
The local evaluations presented in Figure 6 reveal an overestimation of the AOD for stations
close to dust sources such as Tamanrasset ( Fig. 6a), Zinder ( Fig. 6b) and Banizoumbou
( Fig. 6c). This overestimation is more limited with increasing distance from the dust source
at Cinzana (Fig. 6d) and Dakar (Fig. 6e). The order of magnitude of the dispersion between
the three simulations is small when compared to the errors of the simulation in representing
the observed AOD. As a consequence, the uncertainty associated with the choice of the dust
aerosol initial and boundary condition dataset is limited. Overall, the AERONET AOD
measurements appear to be very scarce, particularly close to the dust aerosol sources
(Zinder, Tamanrasset, Banizoumbou, Cinzana). The AOD measurements are performed by
sun photometers which give recording by pointing at the sun. Thus these recordings are only
available during daytime and with clear sky conditions. In some cases of intense dust
plumes with very high concentration, leading to strong solar radiation absorption, the sun
photometers are technically limited and cannot produce any record or, sometimes, the
AERONET quality control system removes them (Mueller et al., 2015 ; Giles et al., 2019).
This may be the reason for the scarcity of observations in this case study, which focuses on
an intense dust event, increasing the perceived overestimation of the simulations. To
compensate for this, the AOD estimates from MODIS satellite observations have been
added to Figure 6 to complete the data.
Furthermore, the CAMS reanalysis appears to be a reliable dataset for dust AOD estimation,
as it has no overestimation and has the lowest $nMAE$ for all sites. Although it does not
reproduce the AOD dynamics close to the dust source at Tamanrasset and Zinder, it has the
highest correlation coefficient for the other sites. Nevertheless, this result should be
interpreted with caution, given the limited data available for calculating the dataset
evaluation metrics. More research is needed to substantiate this conclusion.

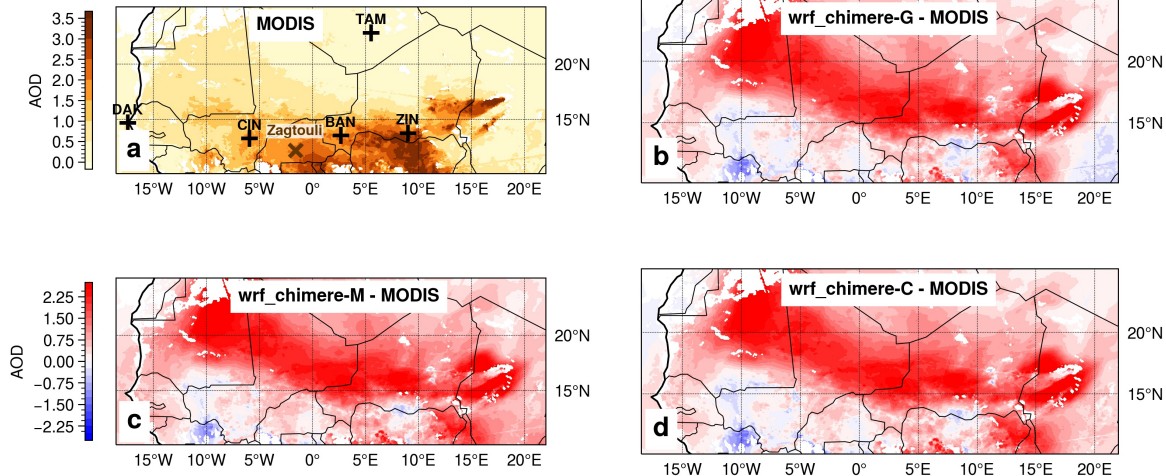

**Figure 7 -** a) Mean from March 28th-00 UTC to April 2nd-00 UTC 2021 of MODIS AOD at
550 nm satellite observations; **x** is the Zagtouli solar farm and **+** corresponds to AERONET
stations. For panels b, c and d, AOD at 550 nm mean differences from March 28th-00 UTC
to April 2nd-00 UTC 2021 between each of the WRF-CHIMERE simulations driven by
GOCART, MERRA2 and CAMS, respectively, and the MODIS satellite observations.
The AOD differences shown in Fig. 7bcd show that the simulations significantly overestimate
the AOD as compared to the MODIS satellite observations, particularly in the Saharan and
North Sahelian zones and in the South Atlas, with an average overestimation of +1.25
between 15°N and 20°N. It is important to note that this overestimation is localised close to
the desert aerosol source zones. The simulated AOD error in the Sahel zone, particularly
around the Zagtouli solar power plant, is more limited with an average of +0.51 between
10°N and 15°N. The mean standard deviation between the three WRF-CHIMERE
simulations is only 10% of the mean error and 5% of the mean simulated AOD.
Consequently the uncertainty in the AOD estimate associated with the selection of the dust
aerosol initial and boundary condition dataset is small.
The observed overestimation of AOD by the WRF-CHIMERE simulations could be due to an
overestimation of the aerosol concentration, or to an inaccurate estimation of the size
distribution of the dust plume, or to excessive aerosol emissions within the domain, or to an
excessive inflow of desert aerosols at the domain boundaries. These hypotheses are
investigated below. Another potential explanation may also be the uncertainties in the
radiative properties of the dust aerosol incorporated in the CHIMERE model, or an
underestimation of the aerosol deposition flux; these aspects are not investigated here.
**3.4.    Aerosol size distribution**

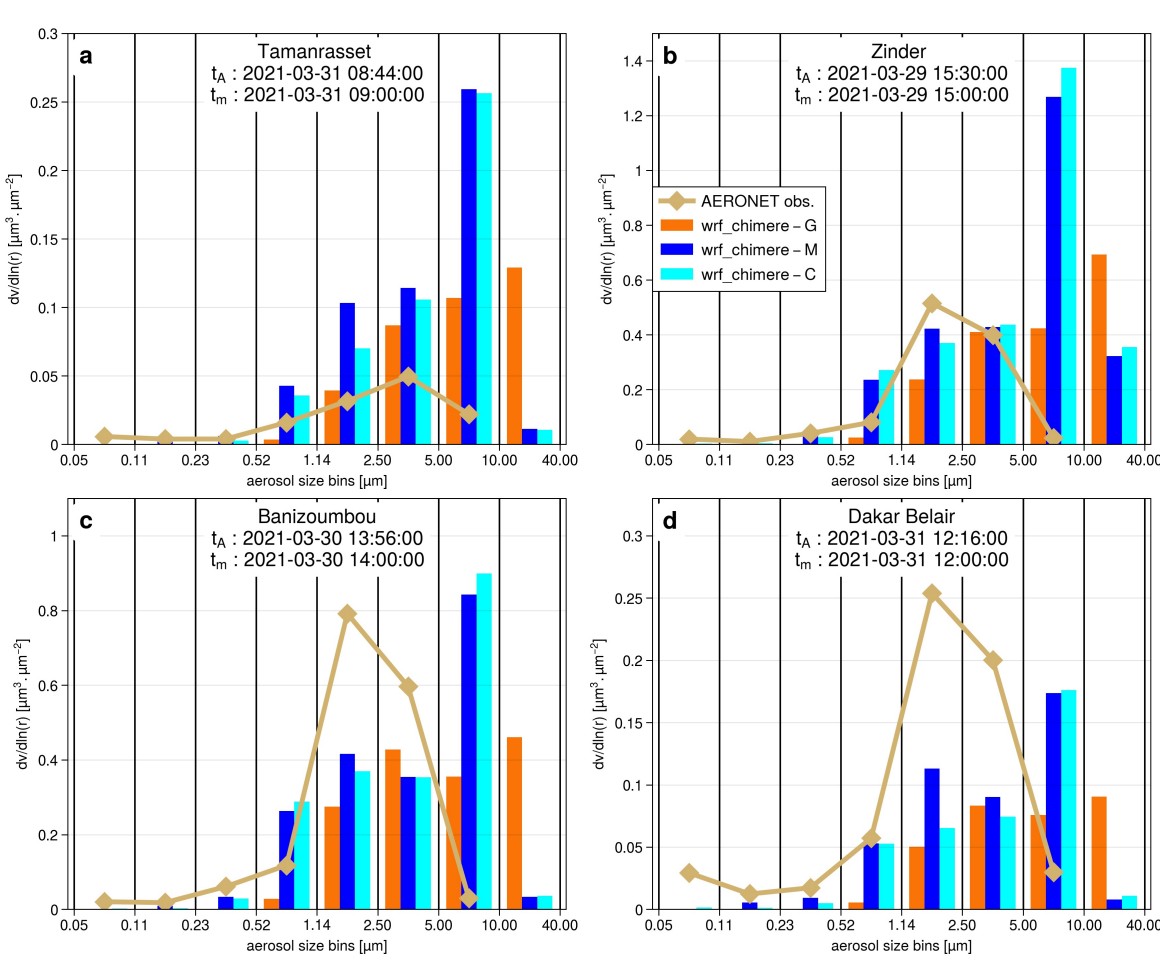

**Figure 8 -** Aerosol volume size distribution for the AERONET station located in a)
Tamanrasset, b) Zinder, c) Banizoumbou and d) DakarBelair. $t_A$ and $t_m$ indicate the times of
the AERONET inversion product and the WRF-CHIMERE model respectively used for the
comparison. *wrf_chimere-G*, *wrf_chimere-M* and *wrf_chimere-C* refer to the WRF-CHIMERE
simulations using GOCART, MERRA2 and CAMS as dust aerosol initial and boundary
condition dataset respectively.
The evaluation of the aerosol size distribution in Fig. 8 shows that the simulations generally
have a dominant aerosol size mode shifted towards coarser sizes compared to the
AERONET inversion product. The ground-based size distribution has a strong peak between
1.14 μm and 5.00 μm, whereas the size distributions estimated by the WRF-CHIMERE
simulations peak for coarser aerosol. For the Dakar Belair station (Fig. 8d), the AERONET
inversion product indicates a first peak of lower intensity between 0.05 and 0.11 μm, which
suggests the presence of aerosols other than desert dust. These aerosols may be of
anthropogenic origin, given the proximity of the measurement site to the Senegalese capital.
When comparing the size distributions between the three simulations with different dust
aerosol initial and boundary condition dataset, it can be seen that the simulations driven with
CAMS and MERRA2 reanalysis are relatively close and well separated from the one driven
with the GOCART climatology. Notably, the dominant size bin in the simulation using
GOCART dataset is consistently the largest particles, whereas with the aerosol from
reanalyses, it is the aerosols between 5 μm and 10 μm. Consequently, the uncertainty
associated with the selection of the dust aerosol initial and boundary condition dataset is
high when examining the aerosol size distribution, particularly for particles exceeding 5.00
μm in diameter. The aforementioned uncertainties in the aerosol size distribution, which are
linked to the choice of the dust aerosol initial and boundary conditions dataset, may be
attributed to differences in the flow of desert dust entering the domain, as well as
uncertainties in the transfer method carried out by the CHIMERE model to match the aerosol
classes of these datasets to its own size distribution, described in section 2.2.3.
As a result, the shift in the WRF-CHIMERE size distribution towards coarser particles
compared to AERONET observations would result in a simulated AOD smaller than
AERONET measurements. However, the opposite is observed (section 3.3). This suggests a
positive bias in the simulated aerosol concentration, which would explain the positive bias in
the AOD, while the coarser size distribution would tend to compensate.
**3.5. Aerosol concentrations**

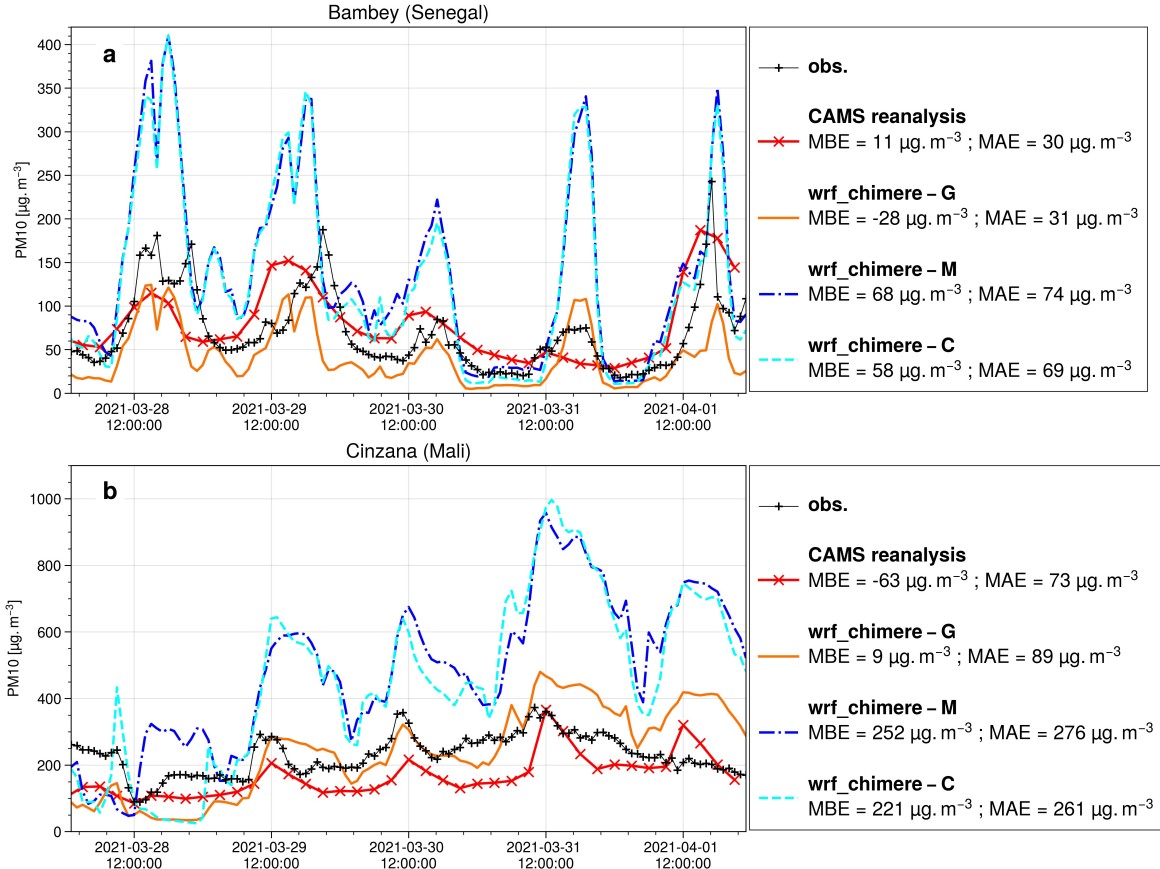

**Figure 9 -** Local comparison of CAMS reanalysis and simulated PM$_{10}$ surface concentrations with INDAAF network observations for a) Cinzana and b) Bambey stations. *wrf_chimere-G*, *wrf_chimere-M* and *wrf_chimere-C* refer to the WRF-CHIMERE simulations using GOCART, MERRA2 and CAMS as dust aerosol initial and boundary condition dataset respectively. MBE is the mean bias error and MAE refers to the mean absolute error.

The three simulations properly capture the dynamics of the PM$_{10}$ surface concentration with respect to the INDAAF ground measurement (Fig. 9) as correlation coefficients are around 0.6 at Cinzana and close to 0.7 at Bambey. The WRF-CHIMERE simulations driven with MERRA2 and CAMS dust aerosol datasets overestimate the surface PM$_{10}$ concentration peaks for Bambey (Fig. 9a) and Cinzana (Fig. 9b), with high positive bias values of around 63 g.m-3 at Bambey and 247 g.m-3 at Cinzana. The latter station is closer to the dust aerosol sources. In contrast, the simulation using the GOCART dust aerosol dataset demonstrates superior performance in representing this variable, with an MAE that is approximately 60% and 70% lower than the two other simulations at Bambey and Cinzana, respectively.

Furthermore, the uncertainty associated with the selection of initial and boundary condition dataset for dust aerosols is of a comparable magnitude to the simulation errors observed for surface PM$_{10}$ concentrations. Section 3.4 partly explains these discrepancies in surface PM$_{10}$ concentration estimates between the simulation driven with the GOCART climatology and those driven with CAMS or MERRA2 reanalysis in terms of aerosol size distribution. These differences may also be attributed to variations in the size distribution of dust aerosol emissions or in the inflow of dust into the simulation domain and its aerosol size distribution.

Furthermore, Fig. 9 indicates that the CAMS reanalysis provides reliable estimates of
surface $PM_{10}$ concentration, as evidenced by the fact it has the lowest MAE values.
However, the Bambey and Cinzana ground measurements, which are the only two available
for the case study, are situated at a considerable distance from the dust sources, limiting our
ability to assess the accuracy of the CAMS reanalysis in capturing the dust event. Moreover,
the CAMS reanalysis exhibits a negative bias at Cinzana, which is the closest site to the dust
sources.

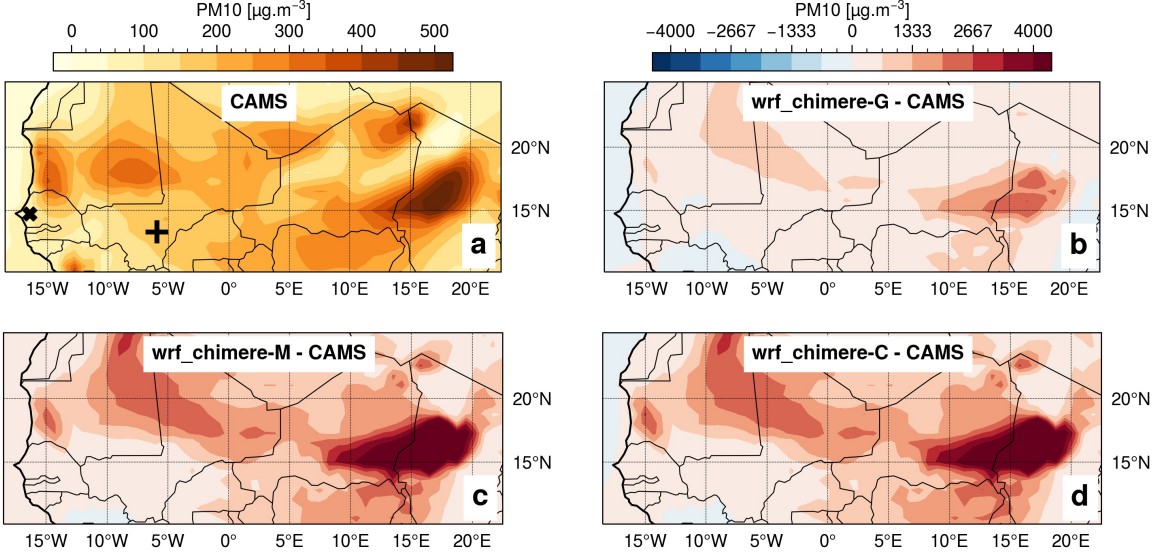

**Figure 10 -** a) Mean from March 28th-00 UTC to April 2nd-00 UTC 2021 of CAMS reanalysis
$PM_{10}$ surface concentration; **x** refers to the Bambey and **+** corresponds to Cinzana INDAAF
stations. For panels b, c and d, $PM_{10}$ surface concentration mean differences from March
28th-00 UTC to April 2nd-00 UTC 2021 between each of the WRF-CHIMERE simulations
driven by GOCART, MERRA2 and CAMS, respectively, and the CAMS reanalysis.
Figure 10 illustrates an overestimation of the $PM_{10}$ concentrations as compared to the CAMS
reanalysis. This is particularly evident in dust source areas such as the Bodélé Depression.
The WRF-CHIMERE simulation driven with the GOCART dataset is the closest to the CAMS
reanalysis, with a mean estimate 3.6 times higher. However, this ratio reaches 8.6 for the
simulations driven with the CAMS and MERRA2 reanalysis dataset.
The mean standard deviation between the three WRF-CHIMERE simulations is 35% of their
mean $PM_{10}$ surface concentration estimate. Consequently the uncertainty in the estimation of
dust $PM_{10}$ surface concentration associated with the selection of the dust aerosol initial and
boundary condition dataset is significant. The discrepancies between the simulation using
the GOCART climatology and the two other ones using CAMS or MERRA2 reanalysis can
be partly explained by the differences in the simulated aerosol size distribution, as shown in
section 3.4.
**3.6.    Dust emissions**

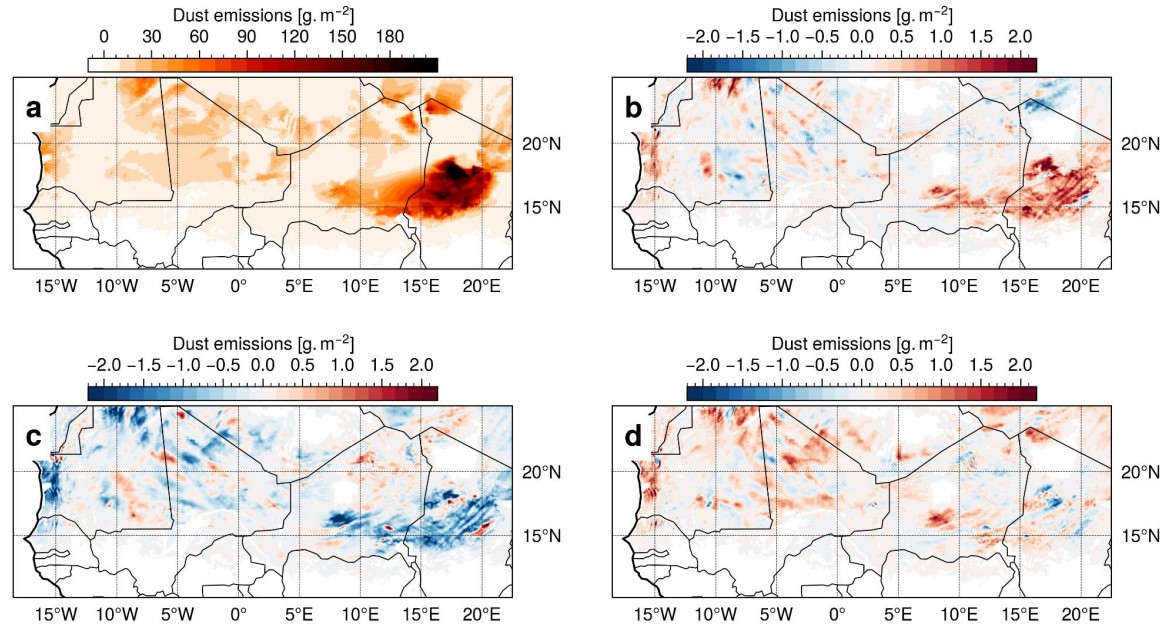


**Figure 11 -** a) Total dust emissions flux from March 28th-00 UTC to April 2nd-00 UTC 2021, averaged between the three WRF-CHIMERE simulations. For panels b, c and d, total dust emissions individual differences between each of the WRF-CHIMERE simulations driven by GOCART, MERRA2 and CAMS, respectively, and the mean of the three WRF-CHIMERE simulations.

In terms of dust emissions (Fig. 11), the Bodélé Depression is, as expected, identified as the primary dust source area, with emissions reaching up to 244 g/m$^2$. The differences of the simulations with each of the three dust aerosol initial and boundary conditions dataset, relative to their mean, exhibit highest values in the source zones located at the Bodélé Depression and the South Atlas. Nevertheless, it is worth noting that there is a factor of 100 in between the emissions in the Bodélé area (approximately 200g/m$^2$) and the observed differences between the three simulations. Consequently, the uncertainties in dust emissions resulting from the choice of the dust aerosol initial and boundary conditions dataset can be considered negligible. As emissions are primarily influenced by surface wind, it can be inferred that the uncertainty generated by the dust aerosol driving dataset on the surface wind is negligible too, which is confirmed by Fig. S4. Additionally, the size distributions of the aerosols emitted during the case study are found to be identical (not shown). Therefore, the differences in dust surface concentration and dust aerosol size distribution may be partly attributed to the dust flows at the boundaries of the domain and are not linked to differences in simulated dust emissions within the domain. However, there is no observational data available to enable a quantitative evaluation of the accuracy of the emissions computed within the WRF-CHIMERE simulations.

### 3.7. Dust boundary flux

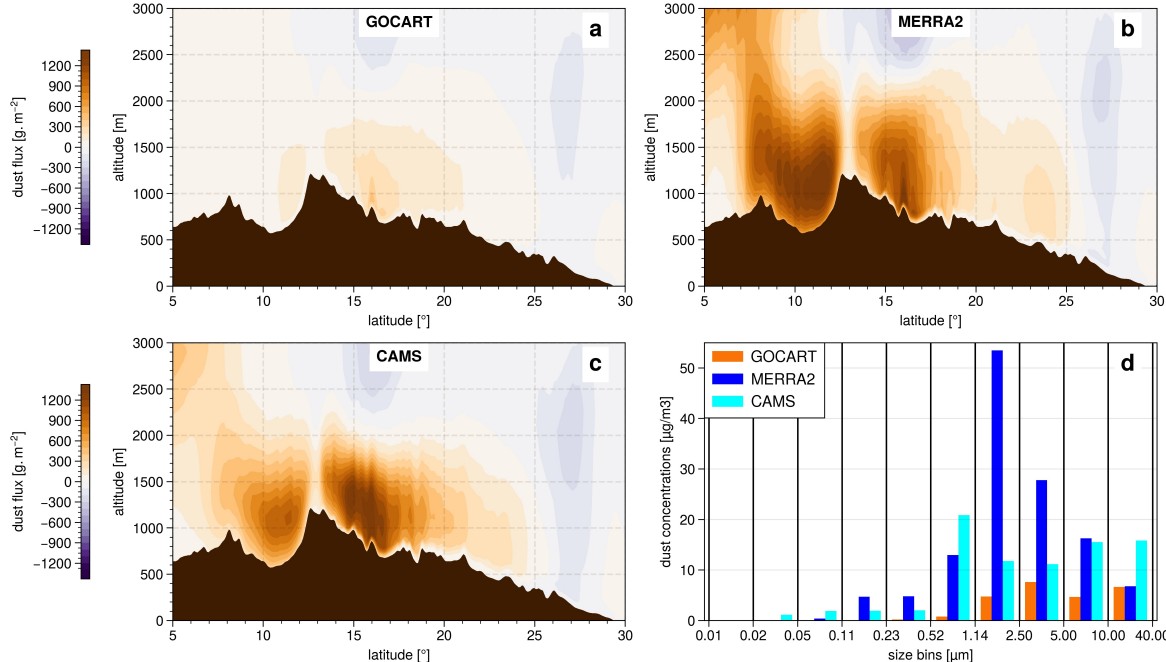

**Figure 12 -** Cumul of the dust flux at the eastern boundary of the simulation from March 28th-00 UTC to April 2nd-00 UTC 2021 for the WRF-CHIMERE simulation with a) GOCART, b) MERRA2 and c) CAMS as dust aerosol initial and boundary conditions dataset; d) Dust size distribution at the eastern boundary limit average during the case study period, from the surface to 200hPa and over latitude. In panel abc, the dust flux is derived as the product between the dust aerosol concentration and the zonal wind, and positive values of the dust flow indicate a flow entering the simulation domain.

As shown in Fig. 1b, the dust event is associated with a strong Harmattan flow, characterised by a northeasterly flow in the lower layer. It is thus interesting to quantify the dust inflow associated with each of the dust aerosol initial and boundary conditions dataset for the eastern domain boundary. The lowest dust flux is observed with GOCART (Fig. 12a), with a maximum of approximately 480 g/m2. In contrast, MERRA2 and CAMS (Fig. 14 b and c respectively) exhibit higher dust fluxes, with maximum values of around 1650 g/m2. The maximum flow is around 10°N for MERRA2, while for CAMS, it is closer to 16°N. Given that GOCART is a climatology, it is reasonable to expect a lower dust flux compared to the CAMS and MERRA2 reanalyses, which are real case simulations incorporating data assimilation of AOD. This is particularly true for the presented case study, which involves an intense dust event associated with a Harmattan flow.

There are also significant differences in both quantity and distribution by aerosol size bin (Fig. 12d). MERRA2 exhibits a strong dominant mode for the class between 1.14 μm and 2.50 μm, while CAMS shows significant values from 0.52 μm to 40 μm, with a maximum for the size class between 0.52 μm and 1.14 μm. Finally, the GOCART model displays a lower variability between 1.14 μm and 40.00 μm, with the maximum occurring for the size class between 2.55 μm and 5.00 μm.

The eastern dust fluxes at the boundary significantly vary depending on the dataset used as dust aerosol initial and boundary conditions, both in terms of quantity and size distribution. The reanalysis dataset, CAMS and MERRA2, are expected to provide a more accurate representation of dust flux in terms of quantity as they are real case simulations assimilating

observational data in their calculations, as compared to GOCART which is a climatology.
However, GOCART provides a more comprehensive description of aerosol size distribution
with seven classes, in comparison to CAMS, which has only three classes but proposes a
higher horizontal resolution. While GOCART considers the effect of aerosol size to be
essential, CAMS assumes the horizontal resolution to be a key parameter. MERRA2 is the
most comprehensive of the three datasets, with the highest horizontal resolution, and an
aerosol size distribution that is close to the GOCART one with five classes.
As a result, and in consideration of the negligible uncertainty in dust emissions within the
simulation domain related to the choice of the dataset for dust aerosol initial and boundary
conditions (see 3.6), these differences in eastern dust fluxes appear to account for the
uncertainties of the simulated surface dust concentrations (see 3.5) and dust aerosol size
distribution (see 3.4).
**3.8.  Discussions**
The evaluation of the simulated GHI at the Zagtouli solar power plant and the Banizoumbou
site (Fig. 2) indicates a significant enhancement in surface solar irradiance estimation when
WRF is coupled with CHIMERE. Specifically, the local MAE is reduced by approximately
75%. This confirms the relevance of incorporating the dust radiative effect with a coupling
approach, in comparison with the operational forecasts currently employed based on
meteorological models alone. During the dry season, dust events similar to the one
presented here, with emissions at Bodélé and then transport of the plume westwards, are
common. This work therefore calls for forecasters in the photovoltaic sector to better account
for the desert dust cycle in their forecast products. This local evaluation also highlights the
potential benefits of using a regional model rather than a global product, as the WRF-
CHIMERE simulations outperform the CAMS gridded solar radiation product with an average
MAE reduced by approximately 38% at the Zagtouli solar farm and by 70% at the
Banizoumbou site, which is closer to dust sources. These discrepancies are corroborated by
the regional comparison presented in Figure 3, which reveals that the mean WRF-CHIMERE
GHI estimate is 5% lower than the CAMS solar radiation dataset. Additionally, the latter does
not exhibit a geographical pattern with lower GHI estimation along the dust plume trajectory,
in contrast to the WRF-CHIMERE simulations. These results confirm those from Sawadogo
et al. (2023) who recently showed that the CAMS reanalysis have low performances in
estimating solar irradiance during high AOD episodes like the one studied here.
Furthermore, the comparison reveals that incorporating dust in the simulation reduces
surface solar irradiance by 18% in this case study. This reduction is notably higher but
remains within the same order of magnitude as previous studies that integrated dust aerosol
information for solar estimation. For example, Masoom et al. (2021) in India and Mostamandi
et al. (2023) in the Arabian Peninsula reported GHI reductions due to dust of approximately
5-10%. This discrepancy underscores the potential variability of the dust impact on solar
irradiance depending on the method used to account for dust effects in the simulations. In
light of the anticipated expansion of PV production in West Africa, this point underscores the
potential consequences of such dust events if they are not accurately predicted.
The evaluation of local surface temperature (Fig. 4) reveals contrasting results regarding the
effectiveness of the coupled approach. It demonstrates an average local MAE reduction of
approximately 10% compared to the WRF-only simulation. However, the main differences
occur mainly at night, when no photovoltaic is produced, as previously observed by Yue et
al. (2010) and Briant et al. (2017). It can be attributed to the opposing radiative forcing
effects of dust aerosols across different wavelength ranges. In the case of longwave, which
corresponds to terrestrial radiation, the presence of dust aerosols has a warming effect.
Conversely, for shortwave, which corresponds to solar radiation, the presence of dust
aerosols induces a cooling effect. Consequently, during night-time when solely terrestrial
radiation is present, there is an increase in surface temperature. During day-time a
competition between the warming effect of terrestrial radiation and the cooling effect of solar
radiation ensues. The net impact is a decrease in surface temperature, indicating that the
effect of solar radiation dominates, with the cooling effect exceeding the warming effect
(Sokolik and Toon, 1999).
The regional evaluation in Fig. 5 confirms these contrasting results and indicates a reduction
of regional MAE by about 14% with the coupling rather than WRF alone. The overestimation
of surface temperature in dusty areas with the coupling, not present in the WRF only
simulation, reveals the dominant aerosol warming effect during night time as compared to
the cooling effect during daytime. These results align with those of Briant et al. (2017), who
estimated dust-induced warming of up to +5°C during nighttime and cooling of approximately
-1°C during daytime in a 2012 dust event in West Africa. These statements strongly depend
on the accuracy of the ERA5 reanalysis which serves as reference. ERA5 integrates data
assimilation of temperature and incorporates aerosol radiative effects through prescribed
monthly climatologies from the GOCART model, but does not dynamically simulate aerosols.
Due to the limited ground measurements in the Saharan region to constrain the reanalysis,
and to the significant biases that can come when considering a coarse climatology for the
radiative effects of aerosols to represent an intense dust event, it is possible that ERA5
underestimates the aerosol effect in dusty areas.
Nevertheless, despite the improvements demonstrated in solar irradiance and surface
temperature estimation, the WRF-CHIMERE simulations exhibit a notable positive bias in
terms of AOD, as evidenced by the local and regional evaluations presented in Figs. 6 and
7. This overestimation cannot be attributed solely to differences in aerosol concentrations, as
the simulations yield markedly disparate surface concentrations of PM10, depending on the
dust aerosol initial and boundary condition dataset chosen (Fig. 10), while this discrepancies
do not appear in the AOD estimates. However, the results from Yahi et al. (2013) and Léon
et al. (2020) emphasized the importance of considering dust plume height when linking
surface PM10 concentrations to AOD. Therefore, differences in the vertical distribution of the
dust plume, not evaluated in this study due to the lack of quantitative observational data,
could account for part of the observed discrepancies between simulated AODs and surface
PM10 concentrations. This excess of aerosol load may be attributed to an overestimation of
emissions within the domain, but this cannot be verified as there is not any such
measurement. The incoming flux of dust in the domain plays a minor role as shown in Fig.
12 where the flux significantly also varies depending on the dust aerosol initial and boundary
condition dataset employed, while these differences are not any more present in the
simulated AOD estimates. Additionally, the underestimation of aerosol deposition, by
sedimentation (not studied in this research) could be at the origin of the overestimation of the
simulated dust loads. Finally, another potential explanation for these AOD biases may be the
inaccuracies in the dust radiative properties incorporated in the CHIMERE model calculation
(see Table S1 and S2). These depend on the mineralogical composition of the desert dust
particles emitted, which are considered uniform in this work. The radiative properties of
aerosols also depend on their granulometry. In the CHIMERE model, dust aerosols are
treated as spherical particles in the calculation of their radiative properties using Mie theory,
which introduces biases. Adbiyi et al. (2023) showed that ellipsoidal dust particles have a
slightly higher mass extinction efficiency compared to spherical particles. As a result,
accounting for ellipsoidal dust aerosols would lead to a slight increase in AOD associated
with a small decrease in GHI. This study further indicates that dust particles with radii
smaller than 20.0 µm are the primary contributors to dust AOD for shortwave radiation, with
the contribution from larger particles being an order of magnitude lower. Therefore, including
particles larger than 40.0 µm in the CHIMERE model would not significantly affect AOD and
GHI estimates. This is corroborated by Mostamandi et al. (2023), who demonstrated that
dust particles with radii smaller than 3 µm are primarily responsible for the reduction in solar
irradiance, while particles larger than 10 µm mainly contribute to dust deposition, which was
not examined in this study.
The uncertainty associated with the choice of the large scale dust aerosol initial and
boundary condition dataset is very low when considering the variables of interest for solar
production, namely GHI and surface temperature (Fig. 3 and 5). This uncertainty is also low
compared to the performance of simulations for AOD estimation (Fig. 7). This result is similar
when examining dust emissions within the domain, which are nearly identical for the three
coupled simulations (Fig. 11). This can be explained by the fact that dust emissions depend
on the cube of surface wind speed (Marticorena and Bergametti, 1995) which present no
significant signature of the selection of the dust aerosol initial and boundary conditions (Fig.
S4). The aerosols emitted within the chosen domain are much greater than those entering,
as the domain accounts for the main source zones. This is why the simulations are not that
sensitive to dust aerosol large-scale dataset employed. The results regarding the uncertainty
associated with the choice of the dust aerosol initial and boundary condition dataset differs
when examining various elements of the dust life cycle. Indeed, aerosol size distributions
vary significantly between the simulation driven with GOCART on one hand, and simulations
driven with CAMS and MERRA2 on the other hand. GOCART climatology over-represents
aerosols larger than 10 µm compared to the CAMS and MERRA2 reanalyses. These
differences partially account for the significant deviation in surface $PM_{10}$ concentration
estimates (Fig. 10), indicating that reanalysis-type datasets result in much higher values, up
to 3 times higher, compared to climatological-type data which is closer to ground
observations. The dust flux entering the domain may also partly explain these differences. In
fact, this flux is very low with GOCART, with values up to 3.5 times lower than CAMS and
MERRA2 (Fig. 12). The size distribution of this incoming aerosol flux is also a determining
factor.
**4.    Conclusion and perspectives**
This study aims to evaluate the ability of the WRF-CHIMERE coupling to simulate GHI
during a typical dust event in the dry season in West Africa. This event is characterised by a
Harmattan flux associated with significant desert dust emissions over the Bodélé
Depression, with the dust plume subsequently transported westward. This work
demonstrates the utility of coupling a meteorological model with a desert aerosol life cycle
model to represent such events, particularly for improving solar forecasts. Indeed, GHI
estimations are markedly enhanced with this approach compared to using a meteorological
model alone with a 75% reduction of local MAE. Nevertheless, the performance of the WRF-
CHIMERE simulations in representing the aerosol load of this event is more controversial.
There is an overall overestimation of AOD and $PM_{10}$ surface concentration by the coupled
model in the North Sahelian-Saharan zone.
This work also aims at investigating whether the performance of the simulations can be
improved by changing the dust aerosol initial and boundary condition dataset, and to
estimate the uncertainty associated with this choice. The results show that this selection has
almost no influence on the estimation of the solar irradiance, surface temperature and AOD.
On the contrary, the choice of the dust aerosol initial and boundary condition dataset has a
significant impact on the surface $PM_{10}$ concentration and the aerosol size distribution.
This work outlines new research perspectives. Firstly, we observe the difficulty of evaluating
simulations in West Africa due to the scarcity of available observations. Establishing a
denser measurement network or conducting observation campaigns, particularly for GHI,
would help research on solar estimation and forecasting in this region. Additionally, the
WRF-CHIMERE simulations demonstrate significant biases in terms of AOD and $PM_{10}$
surface concentration which are not fully explained here. One potential explanation for this is
an overestimation of dust emission, for which no evaluation is possible. Furthermore,
studying aerosol deposition (not conducted in this work) would complement the study of the
desert aerosol life cycle. On the one hand, an underestimation of deposition might be a
contributing factor to the overestimation of the simulated aerosol load. On the other hand,
dust deposition on solar panels affects solar production by masking the available solar
irradiance (soiling effect), and this should be taken into account in forecasting systems to
conduct optimised cleaning operations. Finally, the study focuses on a typical dust event
during the dry season, presenting essentially aerosol-radiation interaction. It could be
beneficial to test such simulation configuration for more complex cases involving cloud
presence. Indeed, the interaction between aerosols and clouds have a significant impact on
solar forecasting by increasing albedo, extending cloud lifespan, and promoting cloud
formation through increased condensation nucleus concentration (indirect aerosol effects).
**Code and data availablitiy**
WRF namelist configuration files, CHIMERE parameter files, Python codes exploited in this
study and GOCART climatology data can be found on the following Zenodo repository:
https://zenodo.org/records/10808476
ERA5 data can be found on the Copernicus Climate Data Store service :
https://cds.climate.copernicus.eu/cdsapp#!/home
CAMS data were downloaded on the Copernicus Atmosphere Data Store service :
https://ads.atmosphere.copernicus.eu/cdsapp#!/home
MERRA2 data can be found on the dedicated platform from NASA :
https://goldsmr5.gesdisc.eosdis.nasa.gov/data/MERRA2/
Data from AMMA ground measurements stations can be accessed from the dedicated
website : https://amma-catch.osug.fr/-jeux-de-donnees-
INDAAF web page allows access to the data : https://indaaf.obs-mip.fr/catalogue/
AERONET data measurements and inversion products are available through the following
link: https://aeronet.gsfc.nasa.gov/
The MODIS satellite observations are available on the "Level-1 and Atmosphere Archive &
Distribution System Distributed Active Archive Center" platform from NASA :
https://ladsweb.modaps.eosdis.nasa.gov/
**Author contributions**
LC, SA, CL conceptualised the study. LC performed the simulations, the analysis and the
editions of the figures. LC, SA, CL, GB, BM, GS, CB, RL and JT discussed the results. LC
wrote the paper
**Competing interest**
The contact author has declared that none of the authors has any competing interests.
**Acknowledgment**
This work has been supported by the NETWAT project (ANR-22-CE03-0011) operated by
the French National Research Agency. To conduct the simulations, this study has benefited
from access to the IPSL-SU (SPIRIT) cluster within the IPSL Mesocentre ESPRI facility,
supported by the CNRS, UPMC, Labex L-IPSL, CNES and Ecole Polytechnique. The
authors want to thank the WRF and CHIMERE developers for giving free access to their
model. We thank the National Aeronautics and Space Agency for the availability of the
MODIS and the MERRA2 data, the European Center for Medium-Range Weather Forecasts
for the availability of the CAMS and ERA5 data and the investigators and staff who maintain
and provide the AERONET, the INDAAF and the AMMA-CATCH observational data. Finally,
we thank the Sonabel company for their contribution.
During the preparation of this work the authors used Deepl Write (Deepl SE) in order to
improve language and readability. After using this tool/service, the authors reviewed and
edited the content as needed and take full responsibility for the content of the publication.

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
