# Peer review of "Solar radiation estimation in West Africa: impact of dust conditions during 2 2021 dry season"

_EGUsphere, 2024_

## Author Comment (AC1)

**RC1: Referee #1**

**1) General comments**

In this study by Clauzel et al. the effect of including desert dust in estimating surface solar radiation using atmospheric modeling (WRF coupled with CHIMERE chemistry-transport model) is investigated during a dust event in March 2021 for West Africa. Their results show the importance of including dust in estimating surface solar radiation. By using three different datasets for dust initial and boundary conditions the authors demonstrate their influence in reproducing surface solar radiation and temperature and aerosol parameters. The objectives of the study are quite straightforward and are addressed by a thorough analysis. The increased accuracy of the surface solar radiation estimates that the authors provide here for West Africa by including the dust effects is of significance for atmospheric modeling the solar energy sector too.

I consider the topic and results of this manuscript to fit the scope of ACP. However, I have some general and major comments (please see below 1-2) which should be addressed prior to publication.

**Author's response :** The authors would like to thank Referee # 1 for his/her encouraging comments and for his/her thorough reading of the article. Responses to comments and corrections to the article are listed below. The line numbers in our response refer to the reviewed manuscript, they are slightly different from the initial version.

1. It is not clear to me in Section 2.2.3 and specifically in Line 241 "$c_i$ is the dust aerosol concentration" and Table 2 "aerosol size distribution" if the concentration of only the dust aerosol component or the total aerosol concentration was used assuming only dust. Please, clarify and discuss this choice in terms of the notable modeled AOD overestimation.

**Author's response :** The method employed in this study is first to check if dust is the dominant aerosol component for this case study and in the region of interest. As shown in Figure S2, dust (yellow pattern) appears as the main contributor. The concentration of only the dust aerosol component is thus considered for the simulations and analysis. Therefore, the modeled AOD overestimation is only due to the dust misrepresentation by the model, in terms of concentration, size distribution or radiative properties as discussed Lines 832-866.

**Author's changes in manuscript :** In Table 2, ''aerosol size distribution'' has been changed to ''dust aerosol size distribution''.

In terms of assimilation, AOD is assimilated. Could you please also provide information regarding CAMS product, which version is used? The one after update of 2023? (https://www.ecmwf.int/en/newsletter/176/news/major-upgrade-cams-forecasts-atmospheric-composition, https://confluence.ecmwf.int/display/COPSRV/Implementation+of+IFS+cycle+48r1+for+CAMS ) If the previous one was used, a DOD underestimation is reported and should be discussed in context of the results.

**Author's response :** CAMS dataset used in this paper has been downloaded in March 2024 and extracted from the latest version available. Thus the CAMS version used is the one after the update of 2023, taking into account the correction on DOD.

**Author's changes in manuscript :** ''The version 48R1 of CAMS is used in this study.'' is added Line 232 and ''v48R1'' has been added in Table 3.

I also recommend checking the DOD to AOD ratio to support the hypothesis in Lines 280-284.

**Author's response :** The DOD to AOD ratio from the CAMS reanalysis dataset is above 80% in this case study and for the region of interest. This ratio was checked at the beginning of the study to support the dust-only hypothesis. This is not shown in the paper as we prefer to show the information from satellite observations (MODIS, CALIOP, Figure S2), considered more reliable than the reanalysis in this region. Nevertheless, we have added a comment on the DOD to AOD ratio.

**Author's changes in manuscript :** ''This hypothesis is also reinforced by the dust optical depth (DOD) to AOD ratio derived from the CAMS reanalysis, which exceeds 80% during this case study and for the domain of interest (not shown).'' was added Line 282.

2. The 3.8 Discussions section needs to be elaborated (see specific comments below) and it lacks comparing the results with the previous literature in the topic. Please, provide (where applicable) relevant references. For example, there are studies showing the direct impact of aerosols in GHI under higher aerosol loads.

**Author's response :** Thanks for this relevant comment that also was given by Referee #2. This point has been corrected (see below).

**Author's changes in manuscript :**

Line 790 ''These results confirm those from Sawadogo et al. (2023) who recently showed that the CAMS reanalysis have low performances in estimating solar irradiance during high AOD episodes like the one studied here'' has been added.

Line 794 ''This reduction is notably higher but remains within the same order of magnitude as previous studies that integrated dust aerosol information for solar estimation. For example, Masoom et al. (2021) in India and Mostamandi et al. (2023) in the Arabian Peninsula reported GHI reductions due to dust of approximately 5-10%. This discrepancy underscores the potential variability of the dust impact on solar irradiance depending on the method used to account for dust effects in the simulations.'' has been added.

Line 805 ''However, the main differences occur mainly at night, when no photovoltaic is produced, as previously observed by Yue et al. (2010) and Briant et al. (2017). It can be attributed to the opposing radiative forcing effects of dust aerosols across different wavelength ranges. In the case of longwave, which correspond to terrestrial radiation, the presence of dust aerosols has a warming effect. Conversely, for shortwave, which correspond to solar radiation, the presence of dust aerosols induces a cooling effect. Consequently, during night-time when solely terrestrial radiation is present, there is an increase in surface temperature. During day-time a competition between the warming effect of terrestrial radiation and the cooling effect of solar radiation ensues. The net impact is a decrease in surface temperature, indicating that the effect of solar radiation dominates, with the cooling effect exceeding the warming effect (Sokolik and Toon, 1999).'' has been added.

Line 821 ''These results align with those of Briant et al. (2017), who estimated dust-induced warming of up to +5°C during nighttime and cooling of approximately -1°C during daytime in a 2012 dust event in West Africa.'' has been added.

Line 874 ''(Marticorena and Bergametti, 1995)'' has been added.

Corresponding references have been added to the reference section.

**2) Specific comments**

**Line 205:** For Table S1 is there any reference for the provided Dust RI?

**Author's response :** The dust refractive indexes given in this table are provided by the CHIMERE model itself. The model documentation refers to Kandler et al., 2007 (Chemical composition and complex refractive index of saharan mineral dust at Izana, Tenerife (Spain) derived by electron microscopy).

**Author's changes in manuscript :** ''given in CHIMERE model (Menut et al., 2021 ; Kandler et al., 2007)'' was added to the Table caption, and the reference Kandler et al. (2007) was added into the References section.

Regarding information in Table S2, could the authors discuss the results in case of larger dust particles than 40µm or non-spherical are considered in the modeling approach?

**Author's response :** The size section range selection is an important choice in the modelling strategy. The selection of 10 bins from 0.01 µm to 40.0 µm was done according to the CHIMERE model documentation (10 bins giving reliable results) and previous studies with CHIMERE model (Briant et al., 2017 ; Menut, 2023). This element has been added to the discussion (see below).

**Author's changes in manuscript :** Line 853 ''The radiative properties of aerosols also depend on their size distribution. In the CHIMERE model, dust aerosols are treated as spherical particles in the calculation of their radiative properties using Mie theory, which introduces biases. Adbiyi et al. (2023) showed that ellipsoidal dust particles have a slightly higher mass extinction efficiency compared to spherical particles. As a result, accounting for ellipsoidal dust aerosols would lead to a slight increase in AOD associated with a small decrease in GHI. This study further indicates that dust particles with radii smaller than 20.0 µm are the primary contributors to dust AOD for shortwave radiation, with the contribution from larger particles being an order of magnitude lower. Therefore, including particles larger than 40.0 µm in the CHIMERE model would not significantly affect AOD and GHI estimates. This is corroborated by Mostamandi et al. (2023), who demonstrated that dust particles with radii smaller than 3 µm are primarily responsible for the reduction in solar irradiance, while particles larger than 10 µm mainly contribute to dust deposition, which was not examined in this study.'' has been added.

**Lines 351-352:** I cannot find the reference in the manuscript, please provide the reference and specify also if this negative bias is for clear-sky or all-sky CAMS radiation service products?

**Author's response :** Thanks for this comment, the reference was forgotten and this point is now corrected. The negative bias is observed for all-sky CAMS radiation service products.

**Author's changes in manuscript :** The reference Lefèvre 2022 for the CAMS solar radiation validation report has been added in the reference section. '' for all-sky solar irradiance estimates'' has been added Line 354.

**Lines 554-557:** Please provide references for this statement and elaborate.

**Author's response :** The AERONET reference cited in section 2.4.3 gives all the information. Mueller et al., 2015 (Towards Optimal Aerosol Information for the Retrieval of Solar Surface Radiation Using Heliosat) also supports this claim.

**Author's changes in manuscript :** ''(Mueller et al., 2015 ; Giles et al., 2019)'' has been added at the end of this statement Line 577.

**Lines 594-595:** May I miss something here, but I don't get the point of interpolating AERONET lower limit 0.05 to model lower limit 0.01. In addition, I am not seeing anything in Fig. 8 at bins below 0.05 even for the model. Why not interpolate between 0.048-10.0 µm?

**Author's response :** We agree with this comment. The interpolation of AERONET inversion product has been applied between 0.048-10.0 µm and Figure 8 has been changed to show the size sections ranging from 0.048 to 40.0 µm. These corrections help the reading of the figure.

**Author's changes in manuscript :** Line 391 :

''AERONET provides an aerosol size distribution dataset estimated through inversion of the photometers data, as described in Dubovik and King (2000). The algorithm for inversion provides a volume particle size distribution for 22 bins, which are logarithmically distributed for radii between 0.05 µm and 15 µm.''

has been changed to

''AERONET also provides an aerosol size distribution dataset estimated through inversion of the photometers data, as described in Dubovik and King (2000). The algorithm for inversion provides a volume particle size distribution for 22 bins, which are logarithmically distributed for radii between 0.05 µm and 15 µm. For comparison with the modelled aerosol size distribution, this distribution is interpolated on the CHIMERE simulated aerosol size distribution which is composed of 10 bins ranging from 0.01 µm to 40.00 µm in diameter (see Table 1). Given that the coarsest bin (10.00-40.00 µm) is at the limit of the capabilities of the inversion method, and the two thinnest ones (0.010-0.022 µm and 0.022-0.048 µm) are out of the range of the inversion product, the AERONET dataset size sections are interpolated on the CHIMERE size sections ranging from 0.048 to 10.0 µm. Consequently, only comparisons between the three simulations can be made for the three size sections which are out of the range of AERONET product. The column aerosol volume size distribution simulated by the model is calculated for each bin "i" as in Menut et al. (2016) :

$$\frac{dV(r_i)}{d\ln(r_i)} = \sum_{k=1}^{nlevels} \frac{m_{k,r_i} \times \Delta z_k}{\rho_{dust} \times \ln(r_{i,max}/r_{i,min})} \tag{3}$$

where $r_i$ is the mean mass median radius (in µm) and $r_{i,min}$ and $r_{i,max}$ the boundaries of the $i^{th}$ bin. $m_{k,r_i}$ is the dust aerosol mass concentration (the mass of aerosol in one cubic metre of air, in $\mu g.m^{-3}$). $\rho_{dust}$ is the dust aerosol density (the mass of the particle in its own volume,

$\rho_{dust} = 2300 \, kg.m^{-3}$). $\Delta z_k$ is the model layer thickness (in metres), for a total of n levels (here 30 vertical levels).''.

**Lines 597-598:** Could you elaborate this sentence, and explain to the reader how the AOD measurement at 875nm influences here?

**Author's response :** Thank you for this comment, which reveals a lack of clarity in this sentence. ''with a maximum wavelength at which the AOD is measured of 875 nm'' has been removed to avoid any confusion. This information is not of importance here.

**Author's changes in manuscript :** ''with a maximum wavelength at which the AOD is measured of 875 nm'' has been removed. In addition, this paragraph (lines 594 to 608 in the initial manuscript) has been moved to section 2 (lines 391 to 408 in revised manuscript) to answer the above question.

**Lines 763-764:** "MERRA2 dataset might be more accurate" Could this statement be put into perspective regarding the AOD and surface PM10 results?

**Author's response :** On reconsidering this statement, we find that it lacks certainty and seems to be not appropriate here in view of the lack of elements to confirm what is being put forward (lack of observation data, no dedicated assessment).

**Author's changes in manuscript :** ''Despite the absence of observational data that would permit a quantitative evaluation of the eastern dust fluxes, the aforementioned elements suggest that the MERRA2 dataset might be more accurate.'' has been removed.

**Lines 808-811:** I think that a comment and relevant references should be added here regarding comparing AOD which is a total column property with surface PM10 concentrations, where the vertical distribution plays an important role.

**Author's response :** For this case study, no quantitative observational data allow us to evaluate the simulated dust concentration vertical profile. That is why we considered surface PM10 concentration to evaluate the model performances in term of aerosol concentration. However, the results from Yahi et al. (2013) and Léon et al. (2020) highlighted the importance of taking into account the dust plume altitude. Thus, the statement proposed here has been nuanced.

**Author's changes in manuscript :** Initial lines 808-811 have been replaced with lines 835-843 : ''This overestimation cannot be attributed solely to differences in aerosol concentrations, as the simulations yield markedly disparate surface concentrations of PM10, depending on the dust aerosol initial and boundary condition dataset chosen (Fig. 10), while this discrepancies do not appear in the AOD estimates. However, the results from Yahi et al. (2013) and Léon et al. (2020) emphasized the importance of considering dust plume height when linking surface PM10 concentrations to AOD. Therefore, differences in the vertical distribution of the dust plume, not evaluated in this study due to the lack of quantitative observational data, could account for part of the observed discrepancies between simulated AODs and surface PM10 concentrations.''

**Lines 812-813:** I think there is a discrepancy here with the conclusion of Section 3.6 Lines 716-719:

"Therefore, the differences in AOD and dust concentration may be attributed to the dust flows at the boundaries of the domain and are not linked to differences in simulated dust emissions within the domain."

Please, clarify.

**Author's response :** Thank you for your comment. This is an uncorrected typo. In fact, we show that differences in dust flux can partly explain the differences in aerosol concentrations and particle size distribution, but not the results obtained for the simulated AODs.

**Author's changes in manuscript :** Line 722

''Therefore, the differences in AOD and dust concentration may be attributed to the dust flows at the boundaries of the domain and are not linked to differences in simulated dust emissions within the domain.''

has been replaced by

''Therefore, the differences in dust surface concentration and dust aerosol size distribution may be partly attributed to the dust flows at the boundaries of the domain and are not linked to differences in simulated dust emissions within the domain.''.

**Lines 813-816:** Again, I think there is a discrepancy here with Section's 3.7 conclusion:

"... these differences in eastern dust fluxes appear to account for the uncertainties of the simulated aerosol concentrations (see 3.5) and AODs (see 3.3)."

Could you please elaborate on this?

**Author's response :** As for the previous comment, this is an uncorrected typo and ''AODs'' should be replaced by ''dust aerosol size distribution''. Thanks for pointing this error.

**Author's changes in manuscript :** Line 769

''these differences in eastern dust fluxes appear to account for the uncertainties of the simulated aerosol concentrations (see 3.5) and AODs (see 3.3)''

has been replaced by

''these differences in eastern dust fluxes appear to account for the uncertainties of the simulated surface dust concentrations (see 3.5) and dust aerosol size distribution (see 3.4).''.

**3) Technical corrections**

**Figure 6:** For Tamanrasset corrcoef is 0.18? or 0.81?

**Author's response :** This value has been checked and 0.18 is correct. However the correlation coefficient computations do not give a fully reliable result because of the very low

number of data available. This point has been commented to ensure a correct understanding of the figure.

**Author's changes in manuscript :** Line 585 : ''Nevertheless, this result should be interpreted with caution, given the limited data available for calculating the dataset evaluation metrics. More research is needed to substantiate this conclusion.'' has been added.

**Supplementary materials**

**Lines 6-8:** The number of equations needs correction

**Author's response :** this has been corrected.

**Author's changes in manuscript :** the numbering of the equations has been revised and changed to eq. S1,  eq. S2, eq. S3, eq. S4 and eq. S5.

**RC2: Referee #2**

Solar energy forecasting plays a crucial role in energy planning and management. This interesting paper deals with the impacts of AOD on the simulation of temperature and solar irradiance using WRF coupled with CHIMERE over the Sahelian zone in West Africa. The paper is well-written and structured, with a well-explained methodology. The results of this study show that WRF-CHIMERE performs better in simulating GHI and temperature over the studied domain than WRF-only. Moreover, the lateral boundary condition provided by AOD also impacts the output of the WRF-CHIMERE. However, I have a general comment on the paper.

**Author's response :** The authors would like to thanks Referee #2 for his/her thorough reading of the article and his/her encouraging comments. The line numbers in our response refer to the reviewed manuscript.

The comparison between WRF-CHIMERE and WRF-only is not appropriate. The configuration of the WRF-only does not incorporate the optimized configuration for solar energy applications. I suggest the authors use the WRF option with WRF-Solar for this purpose. WRF-Solar has all the features designed for solar energy applications, and some modules were introduced to make this tool robust. For instance, the GHI values from the Fast All-sky Radiation Model for Solar Applications (FARMS) in WRF-Solar are better than the traditional ones in the WRF model (Gueymard et al., 2018). I also suggest using the Thompson microphysics scheme for the WRF-only experiment. I recommend the authors use the recommended configurations in WRF for solar energy applications (https://ral.ucar.edu/solutions/products/wrf-solar).

Gueymard, C. A., and P. A. Jimenez, 2018: Validation of real-time solar irradiance simulations over Kuwait using WRF-Solar. *12th Int. Conf. on Solar Energy for Buildings and Industry*, Rapperswil, Switzerland, International Solar Energy Society, https://doi.org/10.18086/eurosun2018.09.14.

This is optional: the authors can also use AOD in WRF-Solar to compare with WRF-CHIMERE; this would provide interesting results for the region.

**Author's response :**

Thank you for this interesting proposal. The question of the model selection was a key step in this research. However, we prefer to use WRF-only than WRF-Solar for the following reasons.

The main goal of this work is to reproduce a dust event in order to quantify dust impact on solar radiation and, at the end, on the solar production. To do so, we selected the WRF-CHIMERE model because it gives one of the most detailed description of the dust life cycle. It also allows a complete computation of the impact of dust particles on the atmospheric dynamics. To highlight the impact of dust on the solar radiation, we need a reference, meaning a simulation without a fully resolved dust life cycle. To ensure that the difference between WRF-CHIMERE simulations and the reference simulations corresponds to only the incorporation of dust, we selected WRF alone in the exact same version (v3.7.1) and configuration as the one for WRF-CHIMERE. WRF-Solar is not appropriate here since its WRF version is different (v4 and later) and requires different physical parametrization.

Please note that we are currently using WRF-Solar for scientific questions complementary to those addressed in this manuscript.

It is difficult for readers to obtain straightforward information from the plotted figures. I suggest writing down the names of different experiments or observed data from the different panels.

**Author's response :** We agree that the figures should be easier to read with some improvements. Thank you for the suggestions applied to increase the clarity of the figures and the manuscript.

**Author's changes in manuscript :** The name of the sites have been added inside the corresponding panels for Figure 2 and Figure 4. The names of the different simulations have been added inside the corresponding panels and the excess colour bars have been removed for Figures 3, 5, 7, 10 and 12. In Figure 9, the x-axis labels have been improved. In Figure 8 the two smallest size sections have been removed (see Referee #1 comment) and the model time outputs chosen have been added inside the panel.

The discussion section needs enhancement by providing references to support your claims.

**Author's response :** Thanks for this relevant comment that also was given by Referee #1. This point has been corrected (see below).

**Author's changes in manuscript :**

Line 790 ''These results confirm those from Sawadogo et al. (2023) who recently showed that the CAMS reanalysis have low performances in estimating solar irradiance during high AOD episodes like the one studied here'' has been added.

Line 794 ''This reduction is notably higher but remains within the same order of magnitude as previous studies that integrated dust aerosol information for solar estimation. For example, Masoom et al. (2021) in India and Mostamandi et al. (2023) in the Arabian Peninsula reported GHI reductions due to dust of approximately 5-10%. This discrepancy underscores the potential variability of the dust impact on solar irradiance depending on the method used to account for dust effects in the simulations.'' has been added.

Line 805 ''However, the main differences occur mainly at night, when no photovoltaic is produced, as previously observed by Yue et al. (2010) and Briant et al. (2017). It can be attributed to the opposing radiative forcing effects of dust aerosols across different wavelength ranges. In the case of longwave, which correspond to terrestrial radiation, the presence of dust aerosols has a warming effect. Conversely, for shortwave, which correspond to solar radiation, the presence of dust aerosols induces a cooling effect. Consequently, during night-time when solely terrestrial radiation is present, there is an increase in surface temperature. During day-time a competition between the warming effect of terrestrial radiation and the cooling effect of solar radiation ensues. The net impact is a decrease in surface temperature, indicating that the effect of solar radiation dominates, with the cooling effect exceeding the warming effect (Sokolik and Toon, 1999).'' has been added.

Line 821 ''These results align with those of Briant et al. (2017), who estimated dust-induced warming of up to +5°C during nighttime and cooling of approximately -1°C during daytime in a 2012 dust event in West Africa.'' has been added.

Line 874 ''(Marticorena and Bergametti, 1995)'' has been added.

Corresponding references have been added to the reference section.

These are detail comments

Line 52: The citation should be (IEA, 2022).

**Author's changes in manuscript :** the citation has been changed as requested.

Line 66: It is solar irradiance, not solar radiation. There is a difference between them. Please, if you refer to GHI, use solar irradiance, not solar radiation. Change this throughout the manuscript.

**Author's response :** We are grateful for your efforts to enhance the precision of the manuscript's vocabulary.

**Author's changes in manuscript :** ''solar radiation'' has been changed to ''solar irradiance'' throughout the full manuscript where applicable.

Lines 143-144: It is hard to see that on Fig.1a. It should be FS1.

**Author's response :** Indeed the dynamic of the dust plume can be seen on Figure S1 rather than Figure 1a, thanks for this relevant comment.

**Author's changes in manuscript :** ''Fig. 1a'' has been changed to ''Figure S1'' Line 145.

Line 349: You could also add Sawadogo et al., 2024:

Sawadogo, W., Bliefernicht, J., Fersch, B., Salack, S., Guug, S., Diallo, B., ... & Kunstmann, H. (2023). Hourly global horizontal irradiance over West Africa: A case study of one-year satellite-and reanalysis-derived estimates vs. in situ measurements. Renewable Energy, 216, 119066.

**Author's response :** Thanks for this additional and interesting article to complete the statement.

**Author's changes in manuscript :** Sawadogo et al. (2023) has been added Line 351 and in the reference section.

Fig.2: I suggest putting the title of the station for each panel. It will be easier for the reader, without having to read the caption.

**Author's response :** Thanks for this comment, this point has been improved.

**Author's changes in manuscript :** Figure 2 has been changed to have the title of the stations with their coordinates for each panel.

Fig.3: Same comment as in Fig.2. In addition, use one color bar for all of them. It would be nice to have the spatial correlation of different experiments based on the CAMS reference dataset.

**Author's response :** This first point has been improved : the names of the different simulations have been added in the corresponding panel and color bars has been removed.

With regard to your second point, the analysis of spatial correlations with the CAMS solar radiation dataset does not seem so relevant to us. Indeed, such an analysis would have to consider this dataset as a reference, which is not as simple as shown in section 2.4.1 where we showed that this dataset has significant biases in desert areas. It is therefore difficult to propose a regional evaluation of WRF-CHIMERE using the CAMS solar radiation dataset as a reference.

**Author's changes in manuscript :** Figure 3 has been improved for reading.

Line 452: Please check the value of 115 W.m−2.

**Author's response :** This value has been checked and is correct. It corresponds to the differences between the mean GHI estimates from the three WRF-CHIMERE simulations and the WRF-only simulation.

Lines 452-453: These values refer to the WRF-only simulation, right? In this case, this should be clearly stated in the sentence.

**Author's response :** Indeed, the reductions of the mean GHI estimation for WRF-CHIMERE or CAMS solar radiation product are computed with the differences with the WRF-only simulation which does not simulate any dust.

**Author's changes in manuscript :** ''as compared to the WRF-only simulation''  has been added Line 471 to precise this point.

Lines 456-458: This statement needs some references. You can use Sawadogo et al., 2024, where they show that CAMS data has a huge bias under the Harmattan period, hence for dust events over Burkina Faso.

Sawadogo, W., Bliefernicht, J., Fersch, B., Salack, S., Guug, S., Diallo, B., ... & Kunstmann, H. (2023). Hourly global horizontal irradiance over West Africa: A case study of one-year satellite-and reanalysis-derived estimates vs. in situ measurements. Renewable Energy, 216, 119066.

**Author's response :** We agree with the fact that CAMS data overestimates solar irradiance under high dust load and we are grateful for the reference provided by Referee #2. However, we have chosen to discuss this point with the appropriate references in the discussion part of the manuscript (section 3.8) rather than in results section. Thus, this point has been corrected further in the article (see below).

**Author's changes in manuscript :** Line 790 ''These results confirm those from Sawadogo et al. (2023) who recently showed that the CAMS reanalysis have low performances in estimating solar irradiance during high AOD episodes like the one studied here'' has been added.

Fig.4: I suggest that wrf_chimere-G, wrf_chimere-M, and wrf_chimere-C refer to the WRF-CHIMERE simulations using GOCART, MERRA2, and CAMS and should be used throughout the manuscript to be consistent.

**Author's response :** We are reassured and satisfied that the naming of the simulations we proposed has been well understood and has been perceived as clear. This naming  has been checked for all figures and the name of the simulation has been added inside the corresponding panel for figures presenting a map.

Lines 477-478: Why do the simulated temperatures differ among the experiments during nighttime? This needs to be discussed.

Line 485: Please provide a scientific explanation for why the impact of dust aerosols on temperature is particularly pronounced at nighttime.

**Author's response :** Thanks for those two comments as we initially did not comment this point in the manuscript. We have corrected this two remarks on night time temperature differences. However we have found it more appropriate to discuss this point in the discussion section (3.8) rather than in the results. See after the corrections given.

**Author's changes in manuscript :** Yue et al. (2010) added in the references section and the following comment has been added Line 805 (section 3.8 discussion) : « However, the main differences occur mainly at night, when no photovoltaic is produced, as previously observed by Yue et al. (2010) and Briant et al. (2017). It can be attributed to the opposing radiative forcing effects of dust aerosols across different wavelength ranges. In the case of longwave, which correspond to terrestrial radiation, the presence of dust aerosols has a warming effect. Conversely, for shortwave, which correspond to solar radiation, the presence of dust aerosols induces a cooling effect. Consequently, during night-time when solely terrestrial radiation is present, there is an increase in surface temperature. During day-time a competition between the warming effect of terrestrial radiation and the cooling effect of solar radiation ensues. The net impact is a decrease in surface temperature, indicating that the effect of solar radiation dominates, with the cooling effect exceeding the warming effect. ».

Lines 554-557: Please provide some references to back up this claim.

**Author's response :** The AERONET reference cited section 2.4.3 gives all the information. Mueller et al., 2015 (Towards Optimal Aerosol Information for the Retrieval of Solar Surface Radiation Using Heliosat) also supports this claim.

**Author's changes in manuscript :** ''(Mueller et al., 2015 ; Giles et al., 2019)'' has been added at the end of this statement Line 577.

Lines 592-606: This part should be in the methodology section.

**Author's response :** Thank you for this suggestion. This correction makes the article easier to read.

**Author's changes in manuscript :** Initial lines 592-606 ''As presented in section […] a total of n levels (here 30 vertical levels).'' have been moved to section 2.4.3 (lines 391-408 in revised version) with slight modification for the clarity of the reading.

For Figure 8, I would like to know the model time output. This is missing in the manuscript.

**Author's response :** The model timestep output is 1h and thus the model time output consider for the comparison is the closest hour to the AERONET time. This point has been clarified by putting the model time output used in the figure.

**Author's changes in manuscript :** The model time output has been added in the panels of the figure. ''The time indicated corresponds to the time of the AERONET inversion product used for the comparison with the simulated aerosol size distribution.'' has been

replace by '' $t_A$ and $t_m$ indicate the times of the AERONET inversion product and the WRF-CHIMERE model respectively used for the comparison'' in the caption of Figure 8.

Lines 771-773: It is hard to understand this sentence. Please rephrase it.

**Author's response :** We agree this sentence was too long and unclear. It has been rephrased for clarity, dividing the sentence in two.

**Author's changes in manuscript :** ''The evaluation of the GHI at the Zagtouli solar power plant and the Banizoumbou site (Fig. 2) shows a clear improvement in its estimation when WRF is coupled to CHIMERE rather than not as the local MAE is reduced by around 75%.'' has been replace by (Line 774) ''The evaluation of the simulated GHI at the Zagtouli solar power plant and the Banizoumbou site (Fig. 2) indicates a significant enhancement in surface solar irradiance estimation when WRF is coupled with CHIMERE. Specifically, the local MAE is reduced by approximately 75%.''

Lines 779-783: The performance of CAMS in simulating solar irradiance during high AOD episodes is low.

Sawadogo, W., Bliefernicht, J., Fersch, B., Salack, S., Guug, S., Diallo, B., ... & Kunstmann, H. (2023). Hourly global horizontal irradiance over West Africa: A case study of one-year satellite-and reanalysis-derived estimates vs. in situ measurements. Renewable Energy, 216, 119066.

**Author's response :** We agree that the discussion section was lacking of reference. Thanks for this suggestion. We have completed the discussion here, also responding your previous comment for Lines 456-458.

**Author's changes in manuscript :** ''These results confirm those from Sawadogo et al. (2023) who recently showed that the CAMS reanalysis have low performances in estimating solar irradiance during high AOD episodes like the one studied here.'' has been added Line 790.

Lines 801-803: This is not true. ERA5 does not dynamically simulate aerosols but incorporates its radiative effects through prescribed monthly climatologies from the GOCART model.

**Author's response :** The sentence was confusing: we wanted to highlight the fact that ERA5 does not dynamically simulate aerosols. The sentence has been modified to clarify this point.

**Author's changes in manuscript :** ''ERA5 integrates data assimilation but does not consider aerosol information in its calculation'' has been replaced by (Line 824) ''ERA5 integrates data assimilation of temperature and incorporates aerosol radiative effects through prescribed monthly climatologies from the GOCART model, but does not dynamically simulate aerosols''. For consistency and clarity, ''and to the significant biases that can come when considering a coarse climatology for the radiative effects of aerosols to represent an intense dust event'' has been added in the next sentence (Line 828).

---

## Author Response (AR2)

**Referee #2**

The authors have replied all my comments and I am satisfied. However, for my first comment regarding the use of the recommended WRF-Solar in this study remains. I know it is difficult to couple the new version that integrates all new schemes required in the WRF-Solar with the CHIMERE model. I do not want to open a new discussion on that. I rather recommend the authors to put it as limitation of this study and to expend the study on that direction.

**Author's response :**

We would like to thank the reviewer for their positive feedback on our responses to the comments. We concur that utilising WRF-Solar would serve to reinforce the findings on the estimation of the solar resource, thereby complementing the results obtained on the impact of desert dust on solar irradiance by comparing WRF alone and WRF-CHIMERE. However, due to technical limitations, we are unable to include simulations with WRF-Solar in this study. As previously mentioned, our team is currently working with this model. Consequently, we have added a note to our article highlighting the use of WRF instead of WRF-Solar as a limitation, and have proposed continuing this research with WRF-Solar in the future. Please see the details below.

**Editor comments**

Dear Authors,

after considering the reports from two reviewers, and my own evaluation of the manuscript, I am glad to accept your paper for publication in ACP, subjet to minor revisions. In particular, please take in consideration the comment by Reviewer #2 and:
1) highlight in Section 2.2 the suitability of your modelling set up to the analysis;
2) discuss in the Conclusions the limitations in using WRF instead of WRF-Solar.

Best regards,
Marco Gaetani

**Author's response :**
Dear Editors,

We would like to thank you for accepting our article for publication in ACP. In order to comply with the minor reviews requested, we propose the following changes:

1) *''In order to reproduce a dust event during the dry season in West Africa, the WRF-CHIMERE coupled model is selected as it has previously demonstrated favourable performance in similar studies such as those conducted by Briant et al. (2017) and Menut (2023). The technical details of this coupled model are provided below. ''* has been added at the beggining of Section 2.2 (now Line 171), in order to highlight the suitability of the modelling set up employed to the analysis ;

2) *''A further limitation of this study is the use of the WRF meteorological model for the coupling with CHIMERE, rather than the WRF-Solar model (Jimenez et al., 2016), which is an enhanced version of WRF dedicated to solar forecasting. Indeed, WRF-Solar incorporates enhanced algorithms for the computation of solar irradiance, accounting for the direct and indirect effects of aerosols and employing an advanced solar tracking algorithm.*

*This makes it the appropriate version of WRF to use for solar energy research. However, no coupling between WRF-Solar and CHIMERE has yet been implemented, representing an important perspective to expend this work."* has been added in the Conclusions section (now Line 928), in order to highlight the limitations in using WRF instead of WRF-Solar and to give the employment of WRF-Solar as a promising perspective.

Please do not hesitate to contact us should you require any further information.
Best regards,
L. Clauzel and co-authors